# Decoding the centromeric nucleosome through CENP-N

**Satyakrishna Pentakota[1†], Keda Zhou[2†], Charlotte Smith[1], Stefano Maffini[1], Arsen Petrovic[1], Garry P Morgan[3], John R Weir[1‡], Ingrid R Vetter[1], Andrea Musacchio[1,4]\*, Karolin Luger[2,5]\***

[1]Department of Mechanistic Cell Biology, Max-Planck Institute of Molecular Physiology, Dortmund, Germany; [2]Department of Chemistry and Biochemistry, University of Colorado at Boulder, Boulder, United States; [3]Department of MCDB, University of Colorado at Boulder, Boulder, United States; [4]Centre for Medical Biotechnology, Faculty of Biology, University Duisburg-Essen, Essen, Germany; [5]Howard Hughes Medical Institute, Maryland, United States

**Abstract** Centromere protein (CENP) A, a histone H3 variant, is a key epigenetic determinant of chromosome domains known as centromeres. Centromeres nucleate kinetochores, multi-subunit complexes that capture spindle microtubules to promote chromosome segregation during mitosis. Two kinetochore proteins, CENP-C and CENP-N, recognize CENP-A in the context of a rare CENP-A nucleosome. Here, we reveal the structural basis for the exquisite selectivity of CENP-N for centromeres. CENP-N uses charge and space complementarity to decode the L1 loop that is unique to CENP-A. It also engages in extensive interactions with a 15-base pair segment of the distorted nucleosomal DNA double helix, in a position predicted to exclude chromatin remodelling enzymes. Besides CENP-A, stable centromere recruitment of CENP-N requires a coincident interaction with a newly identified binding motif on nucleosome-bound CENP-C. Collectively, our studies clarify how CENP-N and CENP-C decode and stabilize the non-canonical CENP-A nucleosome to enforce epigenetic centromere specification and kinetochore assembly.
DOI: https://doi.org/10.7554/eLife.33442.001

**\*For correspondence:**
andrea.musacchio@mpi-dortmund.mpg.de (AM);
karolin.luger@colorado.edu (KL)

[†]These authors contributed equally to this work

**Present address:** [‡]Friedrich Miescher Laboratory, Tuebingen, Germany

## Introduction

Accurate segregation of chromosomes from a mother cell to its two daughters during cell division is a prerequisite for healthy cell physiology and for the transmission of the genetic information across generations (*Santaguida and Amon, 2015*). Specialized, conserved molecular machinery dedicated to this crucial function has been identified in the majority of eukaryotic organisms studied to date (*Drinnenberg et al., 2016*; *van Hooff et al., 2017*). The purpose of this machinery is to generate stable linkages between chromosomes, the carriers of genetic information, and the mitotic spindle, the microtubule-based structure devoted to the segregation of chromosomes into the daughter cells.

In the last two decades, substantial progress in our understanding of the molecular features of the chromosome segregation apparatus has been made. A crucial role in this process is played by centromeres, specialized chromatin domains whose defining mark in almost all known eukaryotes is the enrichment of centromeric protein A (CENP-A, also known as CenH3), which replaces histone H3 in nucleosomes (*Fukagawa and Earnshaw, 2014*; *Musacchio and Desai, 2017*). The primary function of centromeres is to provide a platform for the assembly of macromolecular complexes known as kinetochores, whose task in turn is the physical capture of microtubules of the mitotic spindle. Kinetochores contain approximately 30 core subunits, normally subdivided in centromere-proximal and microtubule-proximal groups. The microtubule-proximal subunits (outer kinetochore), which are

directly implicated in microtubule binding, are usually denoted as the KMN assembly, from the name of three sub-complexes, the Knl1, Mis12, and Ndc80 complexes (*Musacchio and Desai, 2017*). The centromere-proximal subunits (inner kinetochore), which are also organized in sub-complexes, are collectively identified as the constitutive centromere associated network (CCAN) because they appear to reside at centromeres for the entire cell cycle (*Cheeseman and Desai, 2008*; *Foltz et al., 2006*; *Izuta et al., 2006*; *Obuse et al., 2004*; *Okada et al., 2006*) (*Figure 1A*).

The ability of CENP-A to nucleate kinetochores depends on its incorporation into nucleosomes (CENP-A nucleosomes) with histones H2A, H2B, and H4. In vitro, these interact specifically and selectively with two CCAN components, CENP-C and CENP-N (*Carroll et al., 2010, 2009*; *Guo et al., 2017*; *Guse et al., 2011*; *Hoffmann et al., 2016*; *Klare et al., 2015*; *Nagpal et al., 2015*; *Samejima et al., 2015*; *Weir et al., 2016*). Binding of these proteins to CENP-A nucleosomes has been shown to require two regions where the CENP-A sequence diverges significantly from that of histone H3, the L1 loop and the C-terminal tail (*Carroll et al., 2009*; *Fachinetti et al., 2013*;

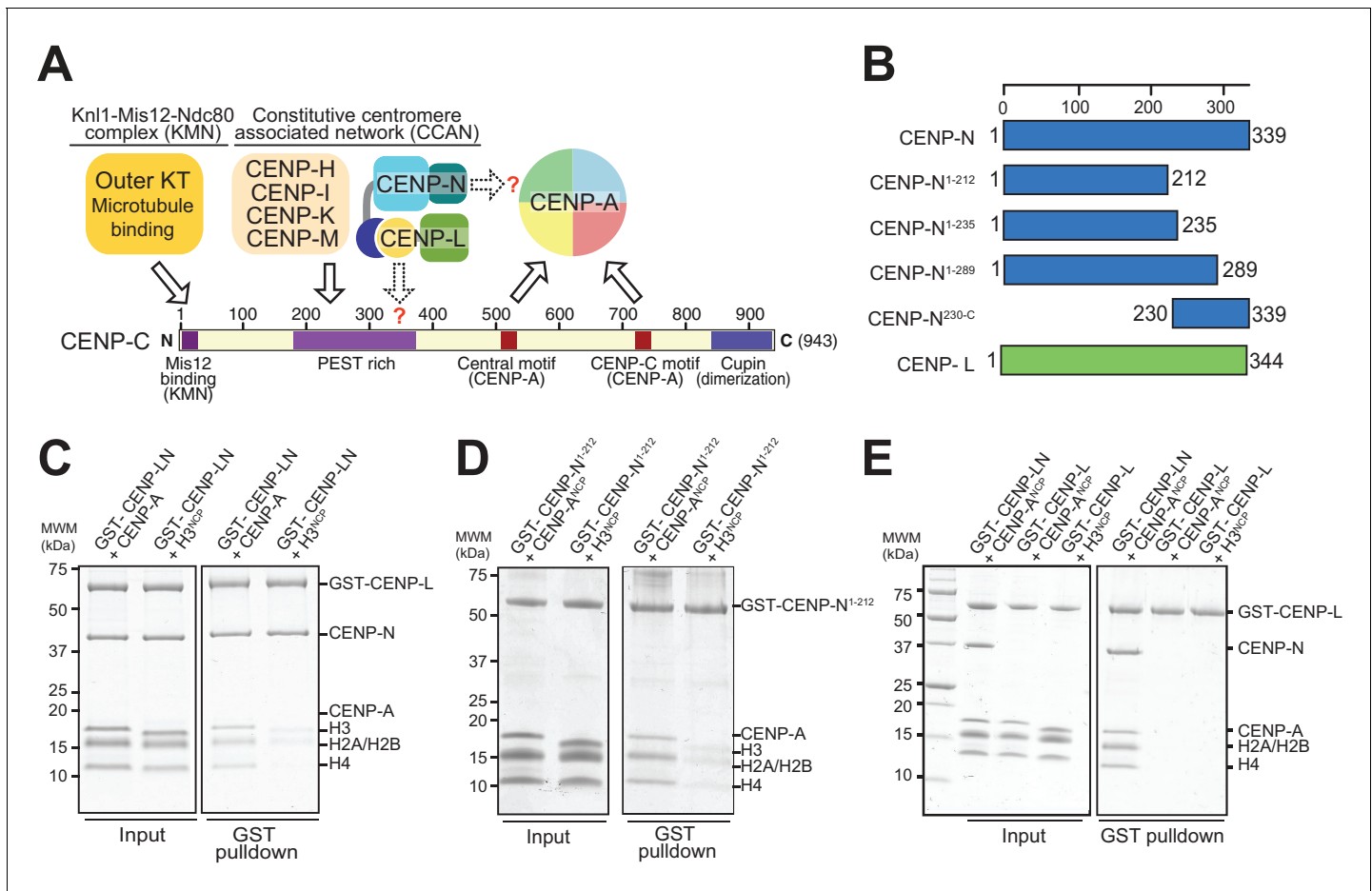

**Figure 1.** The interaction of CENP-N with nucleosomes. (**A**) Schematic of crucial CCAN and KMN subunits discussed in the text. The Knl1-Mis12-Ndc80 (KMN) complex is the main microtubule receptor at the kinetochore. Other interactions are discussed in the main text. The question mark indicates that the precise determinants for the recruitment of CENP-LN to CENP-C and for the interaction of CENP-N with the CENP-A nucleosome have not been identified. (**B**) Schematic depicting constructs described in the manuscript. (**C–E**) Solid phase binding assays where the indicated GST fusion proteins were immobilized on glutathione-sepharose beads (at a final concentration of 1 μM) and incubated with 3 μM of the indicated nucleosome core particles. After incubation (see Materials and methods), beads were centrifuged, washed, and bound proteins visualized by SDS-PAGE and Coomassie staining.

DOI: https://doi.org/10.7554/eLife.33442.002

The following figure supplement is available for figure 1:

**Figure supplement 1.** Further interactions of CENP-NL.
DOI: https://doi.org/10.7554/eLife.33442.003

*Guo et al., 2017*; *Kato et al., 2013*; *Logsdon et al., 2015*). An evolutionary conserved motif of CENP-C, present in one or two copies in different organisms, is sufficient for recognition of CENP-A in vitro. This motif interacts primarily with a solvent-exposed acidic patch on the H2A and H2B subunits of the CENP-A nucleosome and also decodes the divergent C-terminal tail of CENP-A (*Guo et al., 2017*; *Kato et al., 2013*). The two copies of this motif in human CENP-C are referred to as the central motif (or domain) and the CENP-C motif (*Figure 1A*). While at least the central motif has been shown to be required for efficient centromere retention of newly incorporated CENP-A (*Guo et al., 2017*), neither motif appears to be strictly necessary for centromere localization of CENP-C in human cells (*Guo et al., 2017*), likely because CENP-C contains binding sites for additional CCAN subunits that can stabilize its centromere localization even in the absence of a direct interaction with CENP-A (*Guo et al., 2017*; *Hinshaw and Harrison, 2013*; *Klare et al., 2015*; *McKinley et al., 2015*; *Nagpal et al., 2015*; *Weir et al., 2016*). The specific succession of binding sites within CENP-C, a protein that secondary structure prediction algorithms identify as being largely intrinsically disordered, has led to suggest that it acts as a blueprint in the establishment of the inner to outer kinetochore axis, with an N-terminal motif involved in stabilizing the outer kinetochore, a middle region involved in stabilizing the inner kinetochore CCAN complex, and a C-terminal region involved in interactions with the centromeric chromatin (*Figure 1A*) (*Gascoigne et al., 2011*; *Kato et al., 2013*; *Klare et al., 2015*; *McKinley et al., 2015*; *Przewloka et al., 2011*; *Screpanti et al., 2011*).

CENP-N forms a constitutive complex with CENP-L (designated CENP-LN complex), which in turn interacts with the CENP-HIKM complex and with CENP-C (*Guo et al., 2017*; *Hinshaw and Harrison, 2013*; *Klare et al., 2015*; *McKinley et al., 2015*; *Weir et al., 2016*). Binding of CENP-N requires the exposed L1 loop of CENP-A and may also reach into the neighboring DNA (*Carroll et al., 2010*; *Carroll et al., 2009*; *Fang et al., 2015*; *Guo et al., 2017*). The structural basis of the interaction of CENP-N with the CENP-A nucleosome, however, has remained elusive. Furthermore, it is unclear whether this interaction is sufficient for the recruitment of CENP-N to the kinetochore, or whether additional interactions with CCAN subunits are also required. Here, we addressed both issues. First, we combined X-ray crystallography and cryo electron microscopy (EM) to gain a high-resolution view of the CENP-N:CENP-A nucleosome complex, and identified and validated the main determinants of this interaction. Second, we defined the determinants of a physical interaction of CENP-LN with CENP-C and demonstrated that kinetochore recruitment of CENP-N requires the coincident presence of CENP-A and CENP-C at kinetochores. Our studies have important implications for kinetochore assembly and epigenetic specification of centromeres.

## Results

### Crystal structure of CENP-N$^{1-235}$

Human CENP-N, a 339-residue protein (*Figure 1B*), interacts directly with CENP-L (*Hinshaw and Harrison, 2013*; *Weir et al., 2016*). When immobilized on solid phase and challenged with CENP-A or H3 nucleosome core particles (NCPs), CENP-LN interacted specifically with CENP-A$^{NCP}$ (*Figure 1C*). As shown previously (*Carroll et al., 2009*), the CENP-A-binding region of the CENP-LN complex lies within the N-terminal region of CENP-N, because a stable fragment encompassing residues 1–212 of human CENP-N (CENP-N$^{1-212}$) also bound selectively to CENP-A$^{NCPs}$ but not H3$^{NCPs}$ (*Figure 1D*).

To address the structural features of CENP-N and the basis of its interaction with the CENP-A$^{NCP}$, we therefore focused our structural analysis on N-terminal constructs of CENP-N (*Figure 1B*). We obtained well diffracting crystals of the CENP-N$^{1-235}$ construct and determined its crystal structure at 2.8 Å resolution (*Table 1*). CENP-N$^{1-235}$ consists of two closely juxtaposed domains that interact through an extended interface to form a single structural unit (*Figure 2A–B*). The first domain (residues 1–77) consists of a five-helix bundle, whereas the second domain (residues 78–212, cyan in *Figure 1A*) consists of a six-stranded anti-parallel β-sheet sandwiched between α-helices (*Figure 2C–D*). There is no clear density beyond residue ~210, indicating that the structure is disordered after this point. Fold-recognition by DALI (*Holm and Rosenström, 2010*) identified similarity of the first domain to PYRIN domains (PYDs; a superposition is shown in *Figure 2—figure supplement 1A–B*). PYDs are 'death fold' family domains implicated in protein-protein interactions relevant

**Table 1.** X-ray data collection and refinement statistics

**Data collection and processing**

| | Native | SeMet 1 | SeMet 2 | SeMet 1 + 2 |
|---|---|---|---|---|
| Space group | $P4_1$ | $P4_1$ | $P4_1$ | $P4_1$ |
| Wavelength | 0.97793 | 0.9793 | 0.9793 | 0.9793 |
| No. xtals | 1 | 1 | 1 | 2 |
| Source | SLS | PETRA | PETRA | PETRA |
| Detector | Pilatus 6M | Pilatus6M | Pilatus 6M | Pilatus 6M |
| Mol/AU | 2 | 2 | 2 | 2 |
| a,b,c (Å) | 87.3 87.3 81.1 | 88.99 88.99 76.96 | 89.14 89.14 77.22 | 88.99 88.99 76.96 |
| $\alpha, \beta, \gamma$ (°) | 90 90 90 | 90 90 90 | 90 90 90 | 90 90 90 |
| Resolution (Å) | 87.3–2.74 (2.81–2.74)* | 48.7–3.3 (3.9–3.3) | 48.8–3.2 (3.3–3.2) | 48.7–3.3 (3.4–3.3) |
| $R_{meas}$ | 8.2 (155.1) | 17.2 (153.4) | 18.8 (173.4) | 18.7 (167.8) |
| I/σI | 17.3 (1.4) | 7.5 (1.1) | 7.2 (1.0) | 10.4 (1.4) |
| Completeness (%) | 99.8 (98.5) | 100.0 (100.0) | 99.9 (98.8) | 100.0 (100.0) |
| Redundancy | 9.4 (8.7) | 7.1 (7.2) | 7.0 (6.3) | 14.1 (14.1) |
| **Refinement** | | | | **Phasing** |
| Resolution (Å) | 87.3–2.7 | | | FOM 0.39 |
| No. reflections | 17103 | | | BAYES-CC 38.1 |
| $R_{work}/R_{free}$(%) | 21.6/26.1 | | | 12 Selenium-sites |
| **No. atoms:** | | | | |
| Protein/ Ligands | 3432/6 | | | |
| Water | 10 | | | |
| aver. B (Å$^2$) | 90.4 | | | |
| **R.m.s. deviations** | | | | |
| Bond lengths (Å) | 0.0076 | Ramachandran plot: 98.0% favourable, 0% outliers | | |
| Bond angles (°) | 1.27 | | | |

\* Values in parentheses are for highest resolution shell

DOI: https://doi.org/10.7554/eLife.33442.008

to inflammation and apoptosis (*Ratsimandresy et al., 2013*). They have not been previously implicated in interactions with DNA or chromatin.

Iml3 and Chl4 are fungal orthologs of CENP-L and CENP-N, respectively. We referred to a previously reported crystal structure of the full-length Iml3 protein bound to the C-terminal region of Chl4 (Iml3:Chl4[C], PDB ID 4JE3) (*Hinshaw and Harrison, 2013*) to deduce the structural organization of the human CENP-LN complex. Iml3 consists of an N-terminal domain (shown in green in *Figure 2—figure supplement 2A*) and a C-terminal domain (the 'insert' domain shown in yellow; the topology of Iml3 is shown in *Figure 2—figure supplement 2B*). Iml3 hetero-dimerizes with Chl4 through a subdomain within the insert domain (*Figure 2—figure supplement 2B*) (*Hinshaw and Harrison, 2013*). Due to strong sequence similarity of Iml3 and CENP-L throughout their length (not shown), the structure of Iml3 provides an excellent model for the structure of CENP-L. Importantly, although our crystal structure does not encompass the C-terminal region of CENP-N, the sequence of the latter is strongly related to that of Chl4[C] (*Figure 2—figure supplement 3A*), which was captured in complex with Iml3 in the Iml3:Chl4[C] structure, indicating that they are also structurally related (*Figure 2C*). Indeed, as already observed (*Guo et al., 2017*; *Hinshaw and Harrison, 2013*), the C-terminal region of CENP-N (CENP-N[230-C]) was sufficient to interact with CENP-L (*Figure 1—figure supplement 1*). Thus, the structure of CENP-N[1-235] reported here and that of the Iml3:Chl4[C]

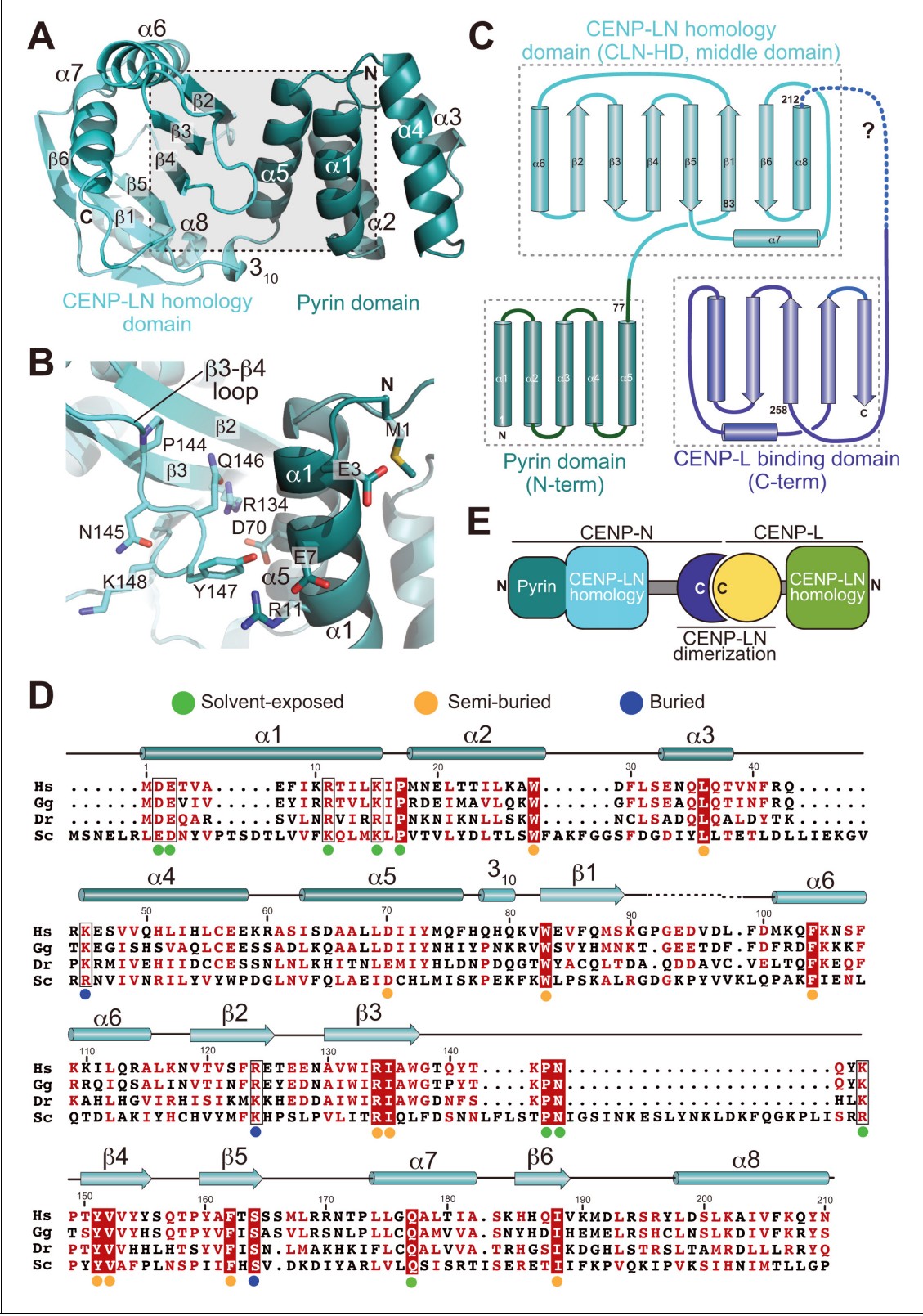

**Figure 2.** Crystal structure of the CENP-A-binding region of CENP-N. (A) Cartoon model of CENP-N[1-235] with secondary structure and domain organization. (B) Close-up of the boxed region in A. (C) Topology diagram of CENP-N. The topology of the Pyrin and CLN-HD domains was directly derived from the crystal structure of CENP-N[1-235] reported here. The topology of the CENP-L-binding domain was derived from the crystal structure of the Chl4 fragment in the complex of the Chl4[CENP-N]:Iml3[CENP-L] yeast homolog (*Hinshaw and Harrison, 2013*). (D) Multiple sequence alignment of

*Figure 2 continued on next page*

*Figure 2 continued*

CENP-N from the indicated species with secondary structure. Green, blue, and orange dots indicate solvent-exposed, semi-buried, and buried side chains, respectively. Positions with conserved residues are displayed red; positions with conserved side chain charge are boxed. (E) Schematic summarizing domain organization of CENP-L, CENP-N, and their dimerization.

DOI: https://doi.org/10.7554/eLife.33442.004

The following figure supplements are available for figure 2:

**Figure supplement 1.** Structural relatedness of CENP-N N-terminal domain with PYRINs.

DOI: https://doi.org/10.7554/eLife.33442.005

**Figure supplement 2.** Topology of Iml3 and CENP-L.

DOI: https://doi.org/10.7554/eLife.33442.006

**Figure supplement 3.** Overall organization of the CENP-LN complex.

DOI: https://doi.org/10.7554/eLife.33442.007

complex are complementary, and together provide an almost comprehensive view of the CENP-L$^{Iml3}$:CENP-N$^{Chl4}$ complex (*Figure 2—figure supplement 3A* and *Figure 2—figure supplement 3B*).

Besides identifying the N-terminal domain of CENP-N$^{1-235}$ as a PYRIN domain, DALI also identified an unanticipated structural homology of the second domain of CENP-N$^{1-235}$ with the N-terminal domain of Iml3$^{CENP-L}$ (*Figure 2—figure supplement 3C*). We therefore refer to these domains of CENP-N and CENP-L as CLN-HD (for CENP-L and CENP-N homology domain). Structural similarities of the CLN-HD suggest that CENP-N and CENP-L are evolutionary related. However, sequence identity of the two domains, even after structural superposition, is minimal, likely explaining why structural similarity had not been predicted (*Figure 2—figure supplement 3D*). CENP-L, or its complex with CENP-N$^{230-C}$, did not interact with CENP-A$^{NCPs}$ or H3$^{NCPs}$ (*Figure 1E* and *Figure 1—figure supplement 1*). Thus, CENP-L and CENP-N, even if partly structurally related, have clearly distinct functions. In conclusion, the structure of CENP-N contains an N-terminal Pyrin domain, a central CLN-HD, and a C-terminal CENP-L dimerization domain, while CENP-L contains an N-terminal CLN-HD, interrupted immediately before the C-terminal helix by an insertion that contains a region required for CENP-N dimerization.

## Cryo-EM analysis of the CENP-N:CENP-A nucleosome complex

Using cryo electron microscopy (cryo-EM), we obtained a three-dimensional reconstruction of CENP-N$^{1-289}$ bound to CENP-A$^{NCPs}$ at ~4.0 Å (*Figure 3A–B*, *Figure 3—figure supplement 1*, and *Table 2*). We built an atomic model of the CENP-N:CENP-A$^{NCP}$ complex by fitting into the EM density high-resolution models of the CENP-A histone core (PDB ID 3AN2) (*Tachiwana et al., 2011*), combined with DNA derived from a nucleosome reconstituted with the 145 bp 601 DNA sequence (PDB ID 3LZ0; *Vasudevan et al., 2010*), and the newly determined crystal structure of CENP-N$^{1-235}$. Both manual and automatic fitting strategies produced unequivocal fits, allowing the first visualization of the interaction of CENP-N with the CENP-A nucleosome (*Figure 3—figure supplement 2*).

The CENP-A nucleosome appears to be stabilized by its interaction with CENP-N (*Guo et al., 2017*). There is clear density for 139 of the 147 bp of DNA and for the N-terminal helix of CENP-A (*Figure 3C* and *Figure 3—figure supplement 2A*), two features reported to be largely disordered and thus invisible in the crystal structure of the CENP-A$^{NCP}$ (PDB ID 3AN2) (*Tachiwana et al., 2011*). CENP-N, whose structure changes very little upon binding to the CENP-A nucleosome, is positioned on top of the L1 loop of CENP-A (also called RG loop for the presence of a conserved arginine-glycine motif at the loop's apex) and contacts approximately 15 base pairs of the adjacent DNA gyre (*Figure 3A*). There is clear density only until CENP-N$^{1-289}$ residue ~210, indicating that the following approximately 80 C-terminal residues (at the opposite end of the nucleosome interaction interface) may be flexible. Of the ~2400 Å$^2$ of CENP-A$^{NCP}$ and CENP-N surface area that become buried in the complex, ~1400 Å$^2$ are at the CENP-N:DNA interface, where both CENP-N domains form extensive interactions with DNA from bp −21 to −35 relative to the twofold axis, or superhelical location [SHL] −2 to −3. There is a marked accumulation of positively charged residues on this DNA binding interface (*Figure 3D*). Four loops in the CLN-HD straddle the DNA double helix over ~8 bp, and the consecutive 7 bp are bound by the PYRIN domain, which is positioned to insert an arginine (R44) into the compressed minor groove in an arrangement that is reminiscent of the minor groove

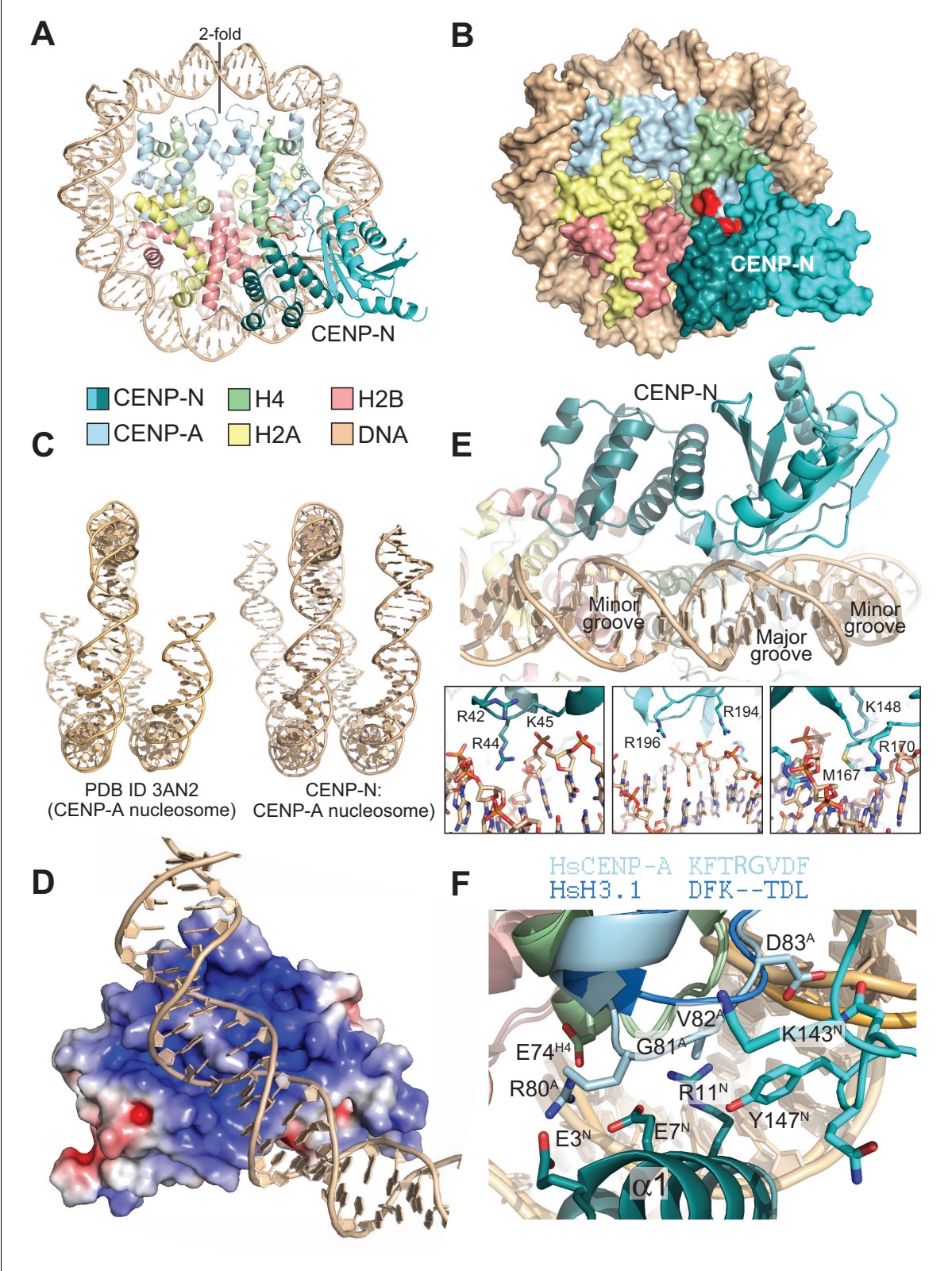

**Figure 3.** The CENP-N:CENP-A$^{NCP}$ complex. (A) Cartoon model of the CENP-A$^{NCP}$ with bound CENP-N$^{1-235}$, determined by cryo-EM. (B) Surface representation of the complex. In A and B, the L1 loop of CENP-A is displayed in red. (C) Comparison of the DNA ends in the crystal structure of the CENP-A nucleosome (*Tachiwana et al., 2011*) and in the structure of the CENP-A:CENP-N complex. (D) Electrostatic potential at the CENP-N DNA binding interface with contour levels $\pm$ 4 $k_B$T/e ($k_B$, Boltzmann constant; T, absolute temperature; e, the magnitude of electron charge, calculated with

*Figure 3 continued on next page*

*Figure 3 continued*

the APBS Pymol plugin). (E) Interaction of CENP-N with backbone, minor groove, and major groove of DNA with close-up views of selected interactions. (F) Interactions at the CENP-A L1 loop and comparison with superimposed H3.

DOI: https://doi.org/10.7554/eLife.33442.009

The following figure supplements are available for figure 3:

**Figure supplement 1.** Additional EM data and analysis.
DOI: https://doi.org/10.7554/eLife.33442.010

**Figure supplement 2.** EM maps.
DOI: https://doi.org/10.7554/eLife.33442.011

**Figure supplement 3.** EMSA assays.
DOI: https://doi.org/10.7554/eLife.33442.012

**Figure supplement 4.** Essential features of CENP-A.
DOI: https://doi.org/10.7554/eLife.33442.013

**Figure supplement 5.** Comparison of CENP-A and H3 and interface with CENP-N.
DOI: https://doi.org/10.7554/eLife.33442.014

**Figure supplement 6.** Comparison of nucleosome binding modes.
DOI: https://doi.org/10.7554/eLife.33442.015

**Table 2.** EM data collection, processing, and refinement statistics

**Data collection and processing**

| | |
|---|---|
| Voltage (kv) | 300 |
| Magnification | 290,000x |
| Defocus (µm, nominal) | −1.0 to −2.5 |
| Pixel size (Å) | 1.02 |
| electron dose rate (counts/pixel/s) | 10 |
| Total electron dose (e$^-$ /Å$^2$) | 80 |
| Exposure time (s) | 8 |
| Number of images (collected/processed) | 3900/3024 |
| Number of frames per image | 40 |
| Initial particle number | 1,843,269 |
| Particle number for 3D classification | 1,267,674 |
| Final particle for refinement | 937,118 |
| Resolution (masked/unmasked) (Å) | 4.0/4.2 |
| Map sharpened b-factor (Å$^2$) | −233 |
| Model refinement | |
| r.m.s. deviation (bonds) | 0.005 |
| r.m.s. deviation (angles) | 0.97 |
| All-atom clashscore | 2.30 |
| Ramachandran plot | |
| Outliers (%) | 0.00 |
| Allowed (%) | 4.59 |
| Favored (%) | 95.81 |
| CaBLAM analysis: | |
| Outliers (%) | 1.92 |
| Disfavored (%) | 6.65 |
| Ca outliers (%) | 0.11 |
| Rotamer outliers (%) | 0.00 |

DOI: https://doi.org/10.7554/eLife.33442.016

arginines inserted by the histones (*Figure 3E*). The highly conserved P17 in the PYRIN domain positions the main chain of CENP-N to latch on to the phosphate backbone of the DNA, with interactions made through the side chains of K15, R42, K45, K81, and R194. There are also likely insertions of CENP-N side chains into two minor grooves (besides R44, also K148, M167, R170) and the intervening major groove at SHL −3 [R196, see also (*Carroll et al., 2009*) (*Figure 3E*)]. In agreement with the presence of a large interaction interface with nucleosomal DNA, CENP-N bound more tightly to CENP-A$^{NCPs}$ but retained substantial binding affinity for H3$^{NCPs}$ in electrophoretic mobility shift assays (EMSAs) (*Figure 3—figure supplement 3A–B*). Likely, this residual binding to H3$^{NCPs}$ in the EMSAs, which emerged less clearly in solid phase binding assays (*Figure 1C*), reflects emphasis on electrostatic interactions under the low-salt conditions of the EMSA assays (150 mM NaCl for complex formation, followed by further dilution upon loading onto the gel), compared to the GST-binding assays (300 mM NaCl), as also discussed in the context of *Figure 4—figure supplement 2*.

## The CENP-N:CENP-A interface

The substantial interface with DNA explains why CENP-N does not bind CENP-A:H4 tetramers lacking DNA (*Carroll et al., 2009*). However, while DNA binding clearly contributes to the binding affinity of this interaction, it is unlikely to contribute to the discrimination of CENP-A$^{NCPs}$ from H3$^{NCPs}$, because CENP-N bound selectively to CENP-A$^{NCPs}$ even when the CENP-A$^{NCPs}$ and H3$^{NCPs}$ contained the same DNA sequence (*Figure 1B–C*). Conversely, the structure clearly suggests why recognition of the L1 loop is crucial for discrimination (*Black et al., 2004*; *Carroll et al., 2010*, *2009*). CENP-N binds the L1 loop through a continuous interface comprising the α1 helix in CENP-N$^{PD}$ and the β3-β4 loop in CENP-N$^{CLN-HD}$. Several of the infrequent conserved solvent-exposed residues of CENP-N (identified by a green dot in *Figure 2D*), including E3, E7, R11, K143, P145, N146, and K148 reside in this interface. Y147, which is less conserved, contributes to the stabilization of the relative arrangements of the CENP-N$^{PD}$ and CENP-N$^{CLN-HD}$, which is largely unchanged in the nucleosome-bound and free structures of CENP-N. Insertion of the side chain of M1 into the hydrophobic core contributes to the stabilization of the α1 helix. The interaction with the CENP-A L1 loop engages a triad of residues, E3$^{CENP-N}$, E7$^{CENP-N}$, and R11$^{CENP-N}$, whose side chains emerge from the same face of the α1 helix looking toward the L1 loop (*Figure 3F*).

The CENP-A residues R80$^{CENP-A}$ and G81$^{CENP-A}$ form a two-residue insertion that is the most conspicuous difference between the L1 loops in CENP-A and H3 (*Figure 3F*, *Figure 3—figure supplement 4A–B*). The insertion is crucial, because it allows R80$^{CENP-A}$ to form hydrogen bonds with both E3$^{CENP-N}$ and E7$^{CENP-N}$, while absence of a side chain at G81$^{CENP-A}$ allows the CENP-A loop to insert deeply into a cleft formed between the two CENP-N domains, where the side chain of Y147$^{CENP-N}$ packs tightly against V82$^{CENP-A}$. In EMSAs, mutation of R80 and G81 to alanine partly ablated the preference of CENP-N for CENP-A$^{NCPs}$ (*Figure 3—figure supplement 3A–B*). The side chain of R11$^{CENP-N}$, a residue previously shown to be important for the CENP-N:CENP-A$^{NCP}$ interaction (*Carroll et al., 2009*), on the other hand, is squeezed between the loop 1 region of CENP-A and the loop 2 region of H4, where it may be involved in a double salt bridge with E74$^{H4}$ and E7$^{CENP-N}$ (*Figure 3F* and *Figure 3—figure supplement 5*).

## Mutational validation of the CENP-N:CENP-A$^{NCP}$ structure

We generated a collection of single and double alanine point mutants to probe the role of individual CENP-N residues in the interaction with the CENP-A$^{NCP}$. In pull-down assays in vitro, we found essentially complete loss of binding with an alanine (A) mutant of R11 (*Figure 4A*), and substantial reductions of binding with alanine mutants of E7 or Y147, at the CENP-A L1 interface, or of K15 or K45, at the interface with DNA (*Figure 4—figure supplement 1A*). Combining mutations of Y147 with either K15 or K45 almost completely disrupted CENP-A$^{NCP}$ binding (*Figure 4A*), in line with the idea that recognition of the L1 loop and of the DNA jointly contribute to the binding affinity of CENP-N for the CENP-A nucleosome. CENP-N targeting to centromeres in U2OS cells reflected the observations made in vitro, with R11A single mutant and the K15A-Y147A and K45A-Y147A double mutants appearing severely impaired in the ability to target centromeres (*Figure 4B–C*), and other single mutants suffering intermediate effects on binding to centromeres (*Figure 4—figure supplement 1B*).

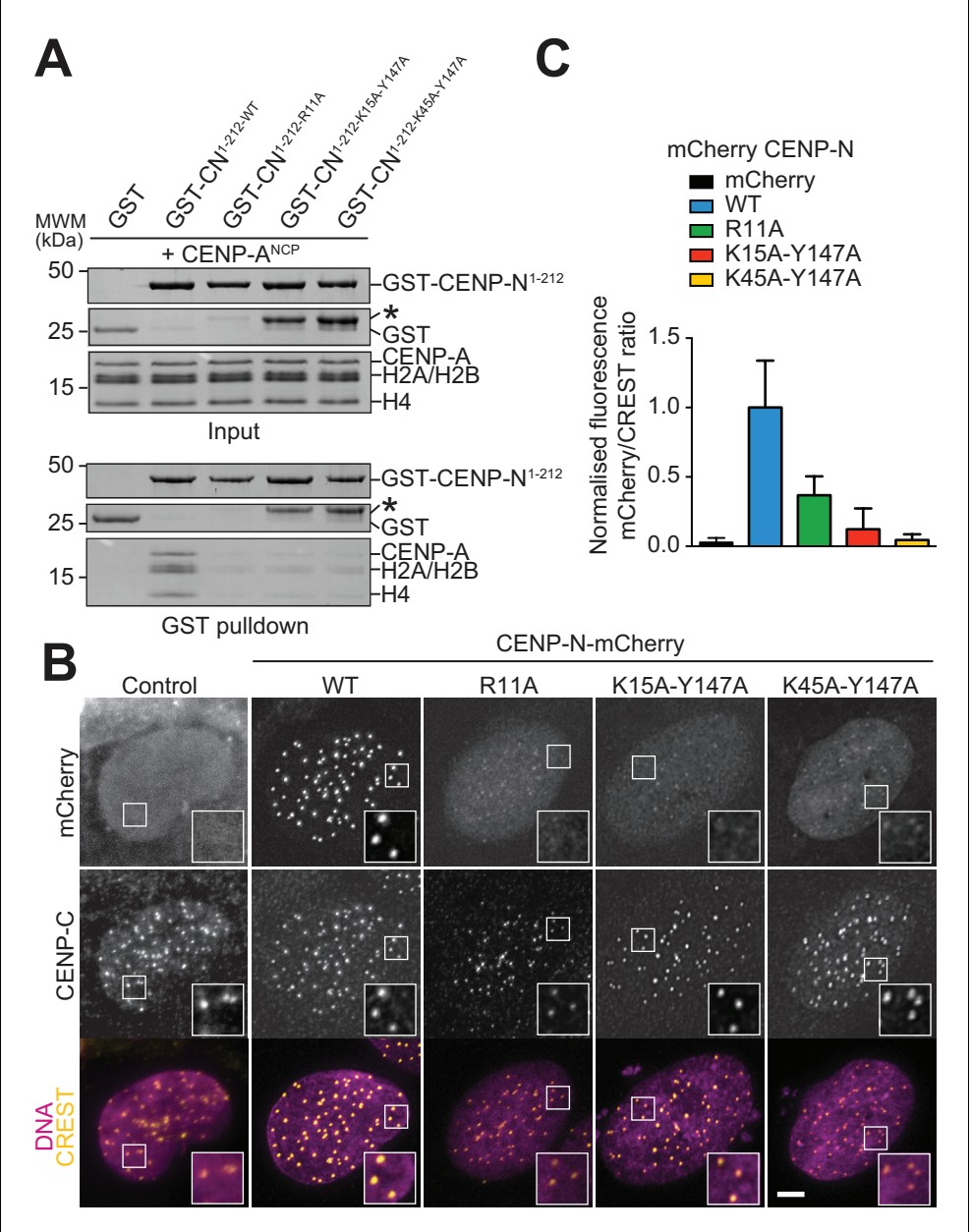

**Figure 4.** Validation of the CENP-N:CENP-A$^{NCP}$ complex. (A) In vitro binding assay probing the interaction of GST-CENP-N$^{1-212}$ immobilized on solid phase with CENP-A$^{NCP}$. (B) Fluorescence microscopy analysis comparing localization at human kinetochores (U2OS osteosarcoma cells) of a wild-type CENP-N-mCherry fluorescent reporter and of its mutant variants. (C) Quantification of localization of the mCherry constructs in B normalized to CREST. The same concentrations of transiently transfected plasmids were compared. Error bars represent SD.
DOI: https://doi.org/10.7554/eLife.33442.017

The following figure supplements are available for figure 4:

**Figure supplement 1.** Characterization of CENP-N mutants in solid phase and cell assays.
DOI: https://doi.org/10.7554/eLife.33442.018

**Figure supplement 2.** Characterization of CENP-N mutants in competition gel shift assays.
DOI: https://doi.org/10.7554/eLife.33442.019

In competition gel shift assays, CENP-N mutants at the CENP-A L1 loop interface (L1 mutants), including R11A and two double mutants (E3A-E7A and K143A-Y147A), lost the ability of CENP-N$^{wt}$ to discriminate between CENP-A and H3-nucleosomes (*Figure 4—figure supplement 2A-B*). We have shown in *Figure 4A* that CENP-N$^{R11A}$ does not bind CENP-A$^{NCPs}$ in solid phase-binding assays. The residual interaction of this mutant with CENP-A$^{NCPs}$ or H3$^{NCPs}$ in EMSAs likely reflects the extensive binding interface of CENP-N for nucleosomal DNA (whose effects are emphasized under low-salt conditions, as already indicated for CENP-N$^{wt}$ in the context of *Figure 3—figure supplement 3*). In line with this interpretation, we find that in the EMSAs the L1 mutants of CENP-N bind H3$^{NCPs}$ indistinguishably from CENP-N$^{wt}$, whereas the same mutants bind to CENP-A$^{NCPs}$ considerably worse than CENP-N$^{wt}$ (*Figure 4—figure supplement 2*). Collectively, these results further emphasize the importance of the L1 loop of CENP-A in selective recognition by CENP-N.

## Identification of a CENP-C region involved in CENP-LN binding

As discussed in the Introduction, CENP-C, an intrinsically disordered protein, provides a defined succession of binding sites for different kinetochore components (*Klare et al., 2015*) (*Figure 1A*). CENP-C, CENP-LN, and CENP-HIKM, another kinetochore sub-complex located in the vicinity of CENP-A, form a 7-subunit complex (designated CHIKMLN) that binds CENP-A$^{NCPs}$ cooperatively, that is with increased binding affinity in comparison to any of the individual subunits or sub-complexes (*McKinley et al., 2015*; *Weir et al., 2016*). Within this assembly, CENP-C$^{2-545}$ binds the CENP-LN complex in vitro [*Figure 5—figure supplement 1A* and (*Hinshaw and Harrison, 2013*; *McKinley et al., 2015*; *Nagpal et al., 2015*; *Weir et al., 2016*).

We set out to exploit biochemical reconstitution and our improved structural understanding of the CENP-LN complex to query the importance of this interaction for kinetochore assembly in humans. Trimming of CENP-C$^{2-545}$ identified CENP-C$^{225-364}$ as a minimal CENP-LN interaction domain (*Figure 5A*), in line with a recent study (*Guo et al., 2017*). Neither CENP-L nor CENP-N$^{1-235}$ bound to CENP-C$^{2-545}$ (*Figure 5—figure supplement 1B–C*). However, the CENP-LN$^{230-C}$ dimer bound CENP-C in the absence of nucleosomes (*Figure 5B*). Thus, CENP-C$^{225-364}$ binds at or near the CENP-LN dimer interface, possibly also exploiting structural ordering of these regions upon dimerization. CENP-C$^{225-364}$ contains a handful of conserved residues, some of which were previously shown to mediate an interaction with the CENP-HIKM complex (*Klare et al., 2015*) (*Figure 5C*). We probed an additional conserved linear motif in CENP-C$^{225-364}$ (residues 302–306) for its potential role in CENP-N binding. A 5-alanine mutant of residues 302–306 (identified as CENP-C$^{5A}$) failed to interact with CENP-NL, identifying this region of CENP-C as the CENP-LN binding motif (*Figure 5D* and *Figure 5—figure supplement 2*). Importantly, CENP-C$^{2-545-5A}$ did not interact with CENP-LN, but retained binding to CENP-A$^{NCPs}$ and CENP-HIKM (*Figure 5E–F*). In isothermal titration calorimetry (ITC) experiments, CENP-LN$^{230-C}$ bound CENP-C$^{225-364}$ with a dissociation constant ($K_D$) of 1 μM, but it showed no binding to CENP-C$^{225-364-5A}$ (*Figure 5G–H*).

## CENP-LN binding motif of CENP-C is required for kinetochore recruitment of CENP-N

The availability of a CENP-LN binding mutant of CENP-C gave us an opportunity to ask if the interaction of CENP-LN with CENP-A, besides being necessary, is also sufficient for kinetochore recruitment of CENP-N. For this, we depleted CENP-C by RNAi and replaced it with exogenous wild type (wt) or mutant (5A) copies. Depletion of CENP-C prevented kinetochore localization of CENP-N, showing that nucleosome binding is not sufficient for CENP-N to reach kinetochores at the low cellular concentration of these proteins. Exogenously expressed wild-type CENP-C promoted CENP-N recruitment, while CENP-C$^{5A}$ failed to promote it (*Figure 6A–B*). Thus, the CENP-LN binding site of CENP-C, while not crucial for CENP-C recruitment to kinetochores, is instead crucial for CENP-N recruitment. Overall, these observations indicate that the CENP-LN complex reads the presence of two features of kinetochores, the presence of CENP-A and the presence of CENP-C, both of which are necessary for its efficient recruitment.

## Discussion

The histone H3 variant CENP-A is an essential feature of centromeres and has two main functions. First, it is required for kinetochore assembly through its direct interactions with inner kinetochore

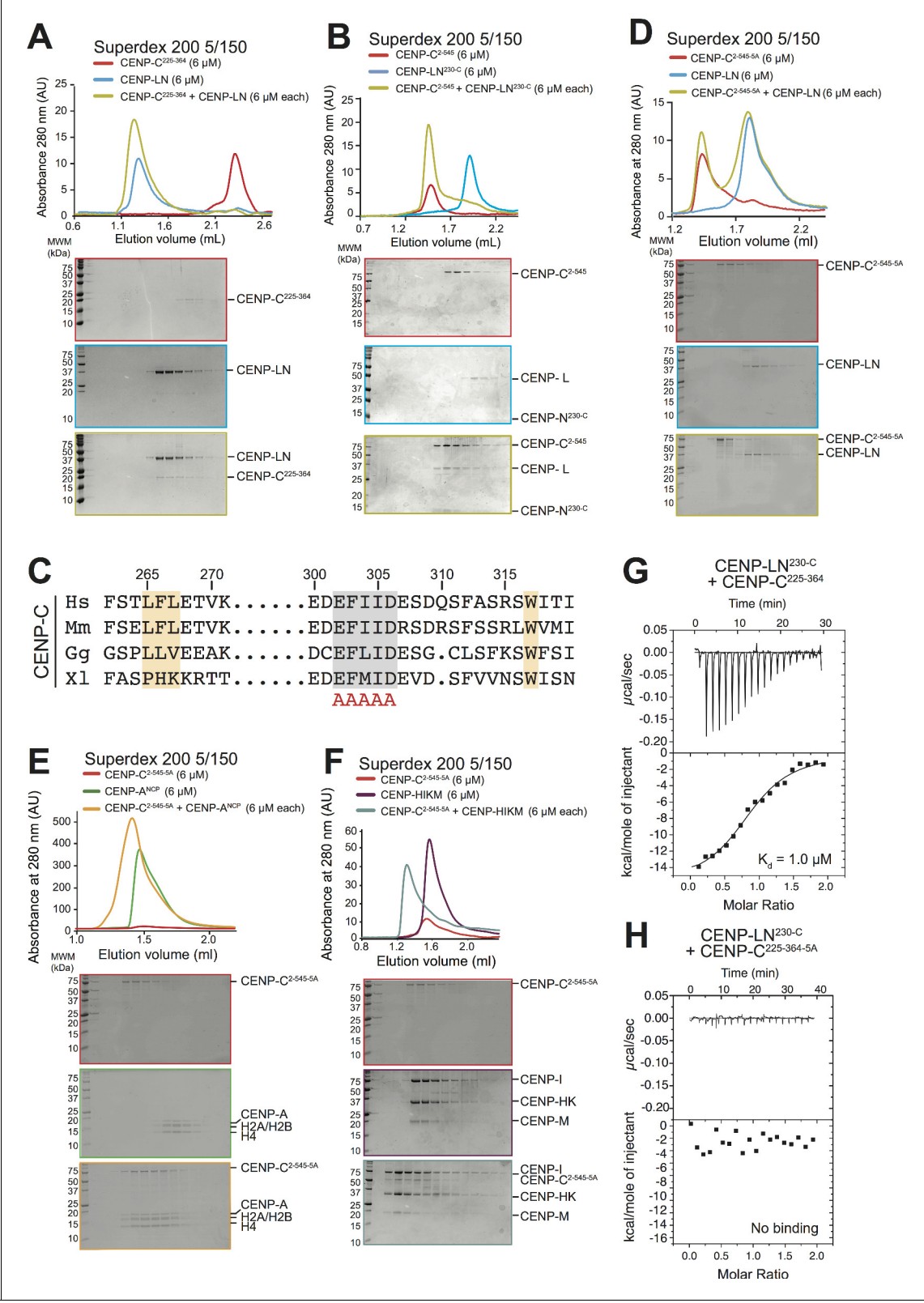

**Figure 5.** Identification of a CENP-N binding site on CENP-C. (**A**) Size exclusion chromatography (SEC) runs of CENP-C$^{225-364}$, CENP-LN complex, and their combination at the indicated loaded concentrations, identified a binding site for CENP-LN in CENP-C$^{225-364}$. Elution fractions were separated by SDS-PAGE and visualized by Coomassie staining. (**B**) CENP-LN$^{230-C}$ binds CENP-C$^{2-545}$, indicating that the CENP-N N-terminal region is not required for CENP-C binding. (**C**) Sequence of a segment of the PEST-rich domain CENP-C that contains a binding site for the CENP-HIKM complex

*Figure 5 continued on next page*

Figure 5 continued

(*Klare et al., 2015*) (residues indicated in salmon). The CENP-N-binding motif is shown in grey. (D) CENP-LN does not bind CENP-C$^{2-545-5A}$. (E) CENP-C$^{2-545-5A}$ retains the ability to bind to CENP-A$^{NCP}$. (F) CENP-C$^{2-545-5A}$ retains the ability to bind to the CENP-HIKM complex. (G) Isothermal titration calorimetry (ITC) experiment quantifying the physical interaction of the CENP-L:CENP-N$^{230-C}$ complex with CENP-C$^{225-364}$. (H) In agreement with the SEC data, CENP-C$^{225-364-5A}$ fails to interact with the CENP-L:CENP-N$^{230-C}$ complex in an ITC experiment.

DOI: https://doi.org/10.7554/eLife.33442.020

The following figure supplements are available for figure 5:

**Figure supplement 1.** Additional size-exclusion chromatography experiments.

DOI: https://doi.org/10.7554/eLife.33442.021

**Figure supplement 2.** Solid-phase-binding assays with CENP-C and CENP-C mutant.

DOI: https://doi.org/10.7554/eLife.33442.022

subunits that can then seed the assembly of this large macromolecular assembly. Second, it is a landmark that determines the stability of centromere chromatin identity through cell division. Interactions of CENP-C and CENP-N with CENP-A$^{NCPs}$ are the only known direct and specific points of contact of the kinetochore with the centromere and are therefore the crucial effectors through which CENP-A implements its role (*Carroll et al., 2010, 2009*; *Kato et al., 2013*).

While the structural basis of CENP-C binding to CENP-A had been described (*Kato et al., 2013*), how CENP-N binds CENP-A had remained elusive. Here, we have filled this important gap, as shown schematically in *Figure 6D–E*. CENP-N nucleosome binding differs from that observed with RCC1 and other nucleosome binders that engage primarily an exposed acidic patch on histones H2A and H2B (*Makde et al., 2010*). However, in its outline it resembles the interaction of the ATPase domain of SWI2/SNF2 chromatin remodeler with H3$^{NCPs}$ (*Farnung et al., 2017*; *Liu et al., 2017*; *Narlikar et al., 2013*), with the important difference that SWI2/SNF2 closely approaches H3 without making significant direct contacts with it, whereas CENP-N interacts directly with CENP-A (*Figure 3—figure supplement 6A–D*). There are also similarities with the nucleosome-binding mechanism of the bromo-adjacent homology (BAH) domain of Sir3 (PDB ID 3TU4) (*Armache et al., 2011*), but the latter interacts predominantly with the H4 N-terminal tail, through recognition of K16$^{H4}$, and with the acidic patch on H2A-H2B, and much less extensively with DNA (*Figure 3—figure supplement 6E–F*). In the CENP-N:CENP-A$^{NCP}$ complex, the normally disordered N-terminal tail of H4 is ordered until R23 and interacts weakly with the CENP-N loop connecting β3 with β4 (*Figure 3—figure supplement 2D*). The reported mono-methylation of K20 of H4 in the CENP-A nucleosome (*Hori et al., 2014*) may further modulate this interaction. In summary, the SWI2/SNF2 and BAH modes of nucleosome binding are predominantly based on interactions with DNA or with the histones, respectively, while CENP-N shows a balance of both. The considerable interaction of CENP-N with DNA is a remarkable and unexpected feature of the complex structure.

CENP-C and CENP-N can interact concomitantly with the same CENP-A nucleosome (*Carroll et al., 2010*), as also confirmed in recent studies (*Guo et al., 2017*; *Weir et al., 2016*). The central motif and the CENP-C motif of CENP-C, which confer CENP-A recognition ability in vitro, interact through a 'arginine anchor' with the acidic patch of H2A and H2B, and also decode the divergent C-terminal tail of CENP-A (*Kato et al., 2013*). These determinants of CENP-C binding on CENP-A are located adjacent to, but not overlapping with, the CENP-N-binding footprint. Indeed, when modeled on the CENP-N:CENP-A$^{NCP}$ structure according to the position it adopts in its structure with the nucleosome (PDB ID 4 × 23) (*Kato et al., 2013*), the CENP-C motif can be accommodated without steric clashes (*Figure 6C,E*). Thus, CENP-C and CENP-N interact with CENP-A through complementary interfaces. In the context of a larger CCAN complex, these CENP-A-binding motifs cooperate to increase the overall binding affinity for CENP-A (*Guo et al., 2017*; *Weir et al., 2016*).

It has been proposed that CENP-N is significantly stabilized upon binding to the CENP-A nucleosome (*Guo et al., 2017*). Our study did not identify a clear structural basis for this phenomenon, as we failed to identify significant conformational changes in CENP-N in isolation (crystal structure) compared to its complex with CENP-A$^{NCP}$. It has also been proposed that CENP-C reshapes and rigidifies the CENP-A nucleosome and that it modulates the DNA termini to make them match the loose wrap observed at centromeres (*Falk et al., 2015*). Importantly, these effects of CENP-C binding on the CENP-A nucleosome do not appear to be required for the selective (over H3) interaction

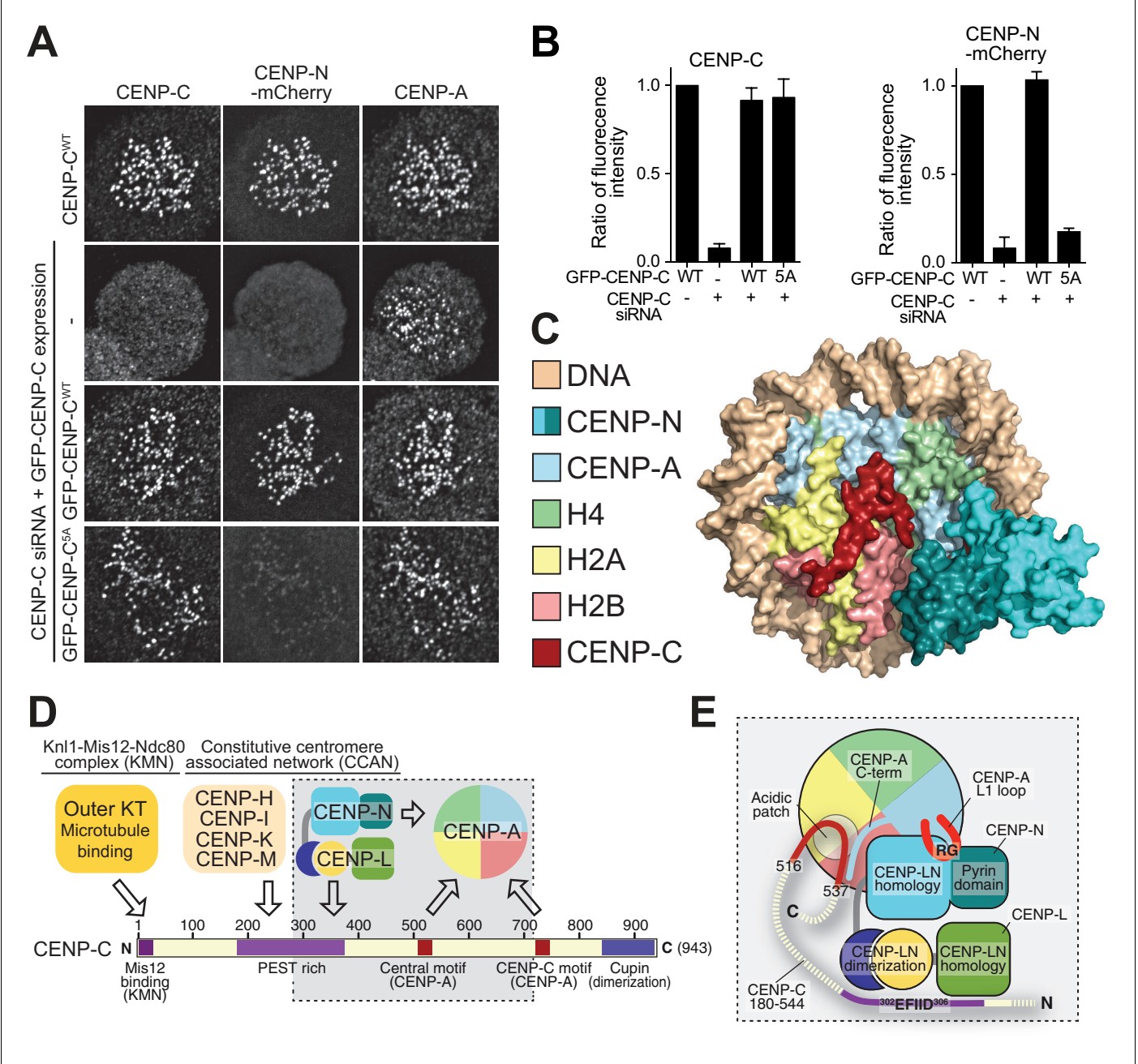

**Figure 6.** Effective CENP-N localization requires CENP-C. (**A**) Fluorescence microscopy analysis comparing kinetochore localization of a wild-type CENP-N-mCherry fluorescent reporter in human HeLa FlpIn TRex cells depleted of CENP-C and, were indicated, further expressing wild-type GFP-CENP-C or the 5A mutant. (**B**) Quantification of CENP-C (left) and mCherry-CENP-N (right) levels at kinetochores in mitotic cells following the rescue of CENP-C depletion by either GFP-CENP-C^{WT} or the GFP-CENP-C^{5A} mutant. Graphs show kinetochore fluorescence intensity of the indicated protein (antibodies against CENP-C or mCherry) normalized to CENP-C or mCherry-CENP-N kinetochore levels in the absence of RNAi treatment, respectively. Each graph is representative of two independent experiments. (**C**) Surface representation of a composite model built by combining the coordinates of the CENP-C motif (residues 712 to 733 from PDB ID 4 × 23, describing its interaction with nucleosome) with those of the CENP-N:CENP-A^{NCP} complex. (**D**) Schematic of crucial kinetochore interactions, already shown in *Figure 1A*, but with question marks removed at interactions investigated in the present work. (**E**) The grey box, an enlargement of the box in D, summarizes the details of the interactions reported in this work, as well as previous information on the interaction of CENP-C with the CENP-A^{NCP}.

DOI: https://doi.org/10.7554/eLife.33442.023

of CENP-N, because selectivity for CENP-A was retained in the absence of CENP-C [this study and (*Carroll et al., 2009*; *Weir et al., 2016*)]. We also note that the DNA termini appear to be well defined in our structure of the CENP-N:CENP-A$^{NCP}$ complex, contrarily to what was observed in the structure of the isolated CENP-A$^{NCP}$ (PDB ID 3AN2). At present, we cannot definitively conclude whether the stabilization of the termini is due to CENP-N binding to the CENP-A nucleosome, as we have not yet been able to obtain a high-resolution EM structure of the CENP-A nucleosome in isolation for comparison. It is possible that the cryogenic conditions used for our structural work stabilize a specific conformation of the complex.

In most organisms, centromere identity is not specified by the centromere's DNA sequence, but rather by the enrichment of CENP-A at a defined chromatin domain. De novo formation of stably inherited centromeres at previously non-centromeric sites (neo-centromeres) provides clear evidence in favor of this idea. Thus, rather than being genetically (i.e. DNA-sequence) specified, centromeres are epigenetically specified, with the pre-existing enrichment of CENP-A being a necessary condition for continued deposition of new CENP-A at the same site through the generations. There is therefore considerable interest in the molecular mechanisms that promote new CENP-A deposition at centromeres during the cell cycle, and in the mechanisms that promote the stabilization and persistence of CENP-A after its incorporation at centromeres.

Conserved machinery for new CENP-A deposition, including the specialized CENP-A chaperone HJURP (Scm3 in *S. cerevisiae*) and an adaptor complex consisting of the Mis18 and M18BP1 subunits, has been described in recent years (*Dunleavy et al., 2009*; *Foltz et al., 2009*; *Fujita et al., 2007*; *Hayashi et al., 2004*; *Pidoux et al., 2009*; *Sanchez-Pulido et al., 2009*; *Williams et al., 2009*). Additional machinery, in particular chromatin remodelling enzymes harnessing ATP hydrolysis to evict H3, is likely involved in the reaction but has not been univocally identified. This machinery is recruited to centromeres early during the cell-cycle and is believed to promote the replacement of histone H3 with new CENP-A (*Dunleavy et al., 2011*; *Jansen et al., 2007*; *Schuh et al., 2007*). Likely, the existing CENP-A nucleosome acts as a template in this reaction, as the abundance of CENP-A nucleosomes at a given centromere is, at least in first approximation, constant through subsequent cell divisions (*French et al., 2017*; *Hori et al., 2017*; *Jansen et al., 2007*). This implies that the same number of new CENP-A nucleosomes is incorporated after each cell division as that of originally present CENP-A nucleosomes, suggesting that the deposition machinery targets, for H3 eviction and replacement with CENP-A, an H3 nucleosome that is likely in close proximity of the CENP-A nucleosome (*Musacchio and Desai, 2017*). While the mechanistic details of CENP-A deposition remain partly unclear, there is now substantial evidence that recruitment of the CENP-A deposition machinery requires CENP-C and possibly other CCAN factors (*Dambacher et al., 2012*; *Moree et al., 2011*; *Shono et al., 2015*). While CENP-N has not been directly implicated in CENP-A deposition, our observation that CENP-N occupies a region of the nucleosome required for binding by chromatin remodelling enzymes of the SWI2/SNF2 family suggests that CENP-N may protect centromeric nucleosomes from remodelling and eviction, thereby contributing to its stability. Indeed, both CENP-C and CENP-N contribute to the stabilization of newly incorporated CENP-A in centromeric chromatin (*Guo et al., 2017*). New CENP-N deposition at centromeres occurs in late S phase (*Fang et al., 2015*; *Hellwig et al., 2011*), and may trigger a stabilization of centromere organization required for successful kinetochore assembly.

In summary, our analysis of the mechanisms of the interactions of the CENP-NL complex with CENP-A and CENP-C represents a step forward in the molecular dissection of the almost universally conserved functions of CENP-A in eukaryotes, which are required for accurate chromosome segregation and, more generally, for the success of cell division and the propagation of life.

## Materials and methods

### Production of recombinant proteins in insect cells

GST-CENP-L and its complexes with CENP-N fragments were produced in insect cells. Specifically, the coding sequence expressing 3C cleavable GST-tagged CENP-L was sub-cloned into MCS2 of the MultiBac vector pFL (*Bieniossek et al., 2012*), while the coding sequences of CENP-N or CENP-N$^{230-C}$ were sub-cloned into MCS1 of pFL for co-expression. All constructs were further transformed into EMBacY cells and subsequently transfected into Sf9 cells in order to produce baculovirus.

Baculovirus was amplified in Sf9 cells and used to infect Tnao38 cells. Tnao38 cells expressing GST-CENP-L:CENP-N, GST-CENP-L:CENP-N[230-C], or GST-CENP-L virus were cultured for 72 hr and isolated (*Weir et al., 2016*). Briefly, cells were resuspended in lysis buffer containing 50 mM HEPES (pH 7.5), 300 mM NaCl, 10% glycerol, 4 mM 2-mercaptoethanol, 1 mM $MgCl_2$ pH 7.5 in presence of Benzonase. Resuspended cells were lysed by sonication and centrifuged at 100,000 g at 4°C for 1 hr. Cleared lysates were incubated with pre-equilibrated GSH-Sepharose beads (Amintra, Expedion, San Diego, USA) at 4°C for 2 hr. After extensive washing with lysis buffer, the GST fusion proteins were eluted in lysis buffer supplemented with 20 mM reduced glutathione. The eluted proteins were concentrated in a 10 K Da Amicon-Ultra-15 Centrifugal filter (Millipore, USA) either alone or in the presence of GST-tagged 3C protease. Concentrated proteins were subsequently loaded onto a Superdex 200 16/600 column equilibrated in 20 mM HEPES pH 7.5, 2.5% glycerol, 300 mM NaCl and 1 mM TCEP. The corresponding peak fractions were collected and concentrated in a 10 KDa MWCO concentrator and then flash frozen in liquid N2, and stored at −80°C until further use.

Full length CENP-N was cloned into a pACEbac1 vector. CENP-N[NT] constructs (containing the N-terminal constructs 1–289 and 1–238) were generated by introducing a sequence encoding 6His followed by two stop codons at the designated positions. CENP-N[NT] constructs were expressed in SF21 insect cells (Invitrogen, Thermo Fisher Scientific, USA). Pellets from 1 L of cell culture (~2 million cells/ml) were lysed in 100 ml lysis buffer (50 mM sodium phosphate pH 7.5, 300 mM NaCl, 20 mM imidazole and 10% glycerol). Nickel NTA beads (1.5 ml) were incubated with the lysed cells overnight at 4°C. Beads were washed with wash buffer (20 mM Tris.Cl pH 7.5, 300 mM NaCl, 30 mM imidazole and 10% glycerol), and eluted with elution buffer (20 mM Tris.Cl pH 7.5, 300 mM NaCl, 150 mM imidazole and 10% glycerol). The concentrated eluate was loaded onto an S200 gel filtration column in buffer containing 20 mM Tris.Cl pH 7.5, 300 mM NaCl, 10% glycerol and 1 mM DTT for the final purification.

## Production of recombinant proteins in bacteria

Fragments encoding different constructs of CENP-N were sub-cloned from a cDNA into pST50Tr-DHFRHIS for expression of recombinant C-terminally polyhistidine-tagged products (*Tan et al., 2005*). To produce GST-tagged CENP-N, a GST encoding sequence was sub-cloned in pST50Tr-DHFRHIS in frame with the sequence coding for CENP-N[1-212]. Mutant CENP-N[1-212] constructs were created by site-directed mutagenesis using the QuikChange kit (Agilent Technologies, Boulder, CO). Escherichia coli (DE3) cells harboring vectors expressing CENP-N[1-212]-His or CENP-N[1-235]-His were grown in TB media supplemented with 100 µg ampicillin at 37°C at an $OD_{600}$ of 0.8–1.0. Then the temperature was reduced to 20°C and protein expression was induced with 0.3 mM IPTG for 16 hr. Cells were harvested at 4600 g for 15 min. Bacterial pellets were resuspended in lysis buffer containing 50 mM HEPES (pH 7.5), 300 mM NaCl, 10% glycerol, 2 mM 2-mercaptoethanol, 10 mM $MgCl_2$ and 5 mM imidazole. Resuspended cells were lysed by sonication and cleared by centrifugation at 100,000 g at 4°C for 30 min. The cleared lysate was incubated with cOmplete[TM] his tag beads (Sigma-Aldrich) and incubated at 4°C for 2 hr. After extensive washing, CENP-N[1-212]-His or CENP-N[1-235]-His were eluted in lysis buffer supplemented with 300 mM Imidazole. A 6 ml ResourceS cation exchange column was pre-equilibrated in 15% buffer B (20 mM HEPES, 1 M NaCl, 10% glycerol, 10 mM $MgCl_2$ pH 7.5, 1 mM TCEP) and 85% buffer A (20 mM HEPES, 10% glycerol, 10 mM $MgCl_2$ pH 7.5, 1 mM TCEP). CENP-N[1-212]-His or CENP-N[1-235]-His were diluted with buffer A to reach a final concentration of 150 mM NaCl, loaded onto the ResourceS column (GE Healthcare Life Sciences and eluted with a linear gradient of buffer B from 150 to 1000 mM NaCl. Fractions containing CENP-N[1-212] His or CENP-N[1-235] His were concentrated in a 10 kDa MWCO and loaded onto a Superdex200 16/600 column equilibrated in 20 mM HEPES pH 7.5, 2.5% glycerol, 300 mM NaCl and 1 mM TCEP. The corresponding peak fractions were collected and concentrated in a 10 kDa MWCO concentrator and then flash frozen in liquid N2, and stored at −80°C until further use. The expression and purification procedure was identical for wild-type or mutants GST-CENP-N sequences, except that the concentration of NaCl in the lysis buffer was raised to 500 mM NaCl (instead of 300 mM).

CENP-C fragments encoding CENP-C[2-545] or CENP-C[225-364] were obtained from codon-optimized CENP-C cDNA and subcloned in pGEX-6P-2rbs for 3′ fusion to the sequence encoding GST. Mutant CENP-C constructs were created by site-directed mutagenesis using the QuikChange kit (Agilent Technologies, Boulder, CO). Expression and purification for all CENP-C constructs and mutants was carried out as described (*Klare et al., 2015*). All constructs were sequence-verified.

## Nucleosome reconstitution

Plasmids for the production of *X. laevis* H2A, H2B, H3 and H4 were a gift from D. Rhodes. Plasmids for the production of human CENP-A:H4 histone tetratmer were a gift from A.F. Straight. 145 bp DNA (601-Widom) wrapped around CENP-A or H3 octamers was a gift from C.A. Davey. Purification of CENP-A or H3 containing NCPs were performed as previously described (*Guse et al., 2011*; *Guse et al., 2012*; *Weir et al., 2016*). For cryo-EM studies, nucleosomes were reconstituted from recombinant human H2A, H2B, CENP-A, and H4. Nucleosomes were reconstituted using the salt dialysis method (*Dyer et al., 2004*). '601' DNA (147 bp), H2A:H2B dimer, and CENP-A:H4 (or H3: H4) dimer were mixed at molar ratios of 1 to 2.4 to 2.4 in buffer containing 20 mM Tris.Cl pH 7.5, 2M NaCl, and 1 mM EDTA. Sample was transferred to a dialysis tube (D-tube Dialyzer, EMD Millipore), and dialyzed against RB high (Reconstitution Buffer high: 20 mM Tris.Cl pH7.5, 2 M NaCl, 1 mM EDTA and 1 mM DTT) for 4 hr at 4°C. Gradient dialysis was set up as described (*Dyer et al., 2004*). A peristaltic pump (flow rate 1.5 ml/minute) was used for replacing RB high with RB low (Reconstitution Buffer low: 20 mM Tris.Cl pH7.5, 0.25 M NaCl, 1 mM EDTA and 1 mM DTT) for 18 hr. Finally, the sample was dialyzed against storage buffer (20 mM Tris.Cl pH7.5, 20 mM NaCl, 1 mM EDTA and 1 mM DTT) for 8 hr.

## GST pull-down assays

GST pulldown assays were performed as previously described (*Klare et al., 2015*) with minor modifications. Briefly, pre-blocked GSH Sepharose beads were incubated in pull-down buffer consisting of 10 mM HEPES pH 7.5, 2.5% glycerol, 1 mM TCEP and either 300 mM NaCl for GST-LN-CENP-C assays or 150 mM NaCl for GST-LN-NCPs assays. GST-CENP-LN or GST-CENP-N$^{1-212}$ as baits (at a 1 µM overall concentration) were incubated with either NCPs or CENP-C as prey at a 3 µM concentration. The bait was loaded onto 10 µl pre-blocked beads before adding the prey. The reaction volume was topped up to 40 µl with buffer and incubated at 4°C for 1 hr. Beads were spun down at 500 g for 3 min. The supernatant was removed and beads were washed twice with 200 µl pull-down buffer supplemented with 0.05% Triton-X-100. Supernatant was removed completely, samples boiled in 15 µl Laemmli sample loading buffer and run on a 14% SDS–PAGE gel. Bands were visualized with Coomassie brilliant blue staining. For pre-blocking of GSH sepharose beads, 750 µl of GSH sepharose beads were washed twice with 1 ml washing buffer (20 mM HEPES pH 7.5, 200 mM NaCl) and incubated in 1 ml blocking buffer (20 mM HEPES pH 7.5, 500 mM NaCl, 500 µg/µl BSA) overnight at 4°C with rotation. Beads were washed five times with 1 ml washing buffer and resuspended in 500 µl washing buffer to have a 50/50 slurry of beads and buffer. Pull-down assays were repeated at least three times.

## Analytical SEC analysis

Analytical SEC was performed on either Superdex 200 5/150 or Superose 6 5/150 columns in a buffer containing 20 mM HEPES (pH 7.5), 300 mM NaCl, 2.5% glycerol, 2 mM TCEP, pH 7.5 on an ÄKTAmicro system (GE Healthcare). Proteins were mixed at 6 µM in a total volume of 50 µl, incubated for one hour on ice, spun for 15 min in a bench-top centrifuge before each injection. All samples were eluted under isocratic conditions at 4°C in SEC buffer (20 mM HEPES pH 7.5, 300 mM NaCl, 2.5% glycerol, 2 mM TCEP) at a flow rate of 0.1 ml/min. Elution of proteins was monitored at 280 nm. Fractions (100 µl) were collected and analyzed by SDS–PAGE and Coomassie blue staining.

## CENP-N:CENP-A nucleosome complex preparation

Complexes containing CENP-A nucleosome and CENP-N$^{NT}$ (CENP-N:CENP-A nucleosome complex) were generated by mixing CENP-A nucleosome and CENP-N$^{NT}$ at a molar ratio of 1:3, then dialyzed against 20 mM Tris.Cl pH 7.5, 50 mM NaCl, 1 mM EDTA and 1 mM DTT. The complexes were concentrated to approximately 2.5 µM using a Millipore concentrator (MW 50 kD cut-off).

## EMSA assays

Nucleosomes were mixed with CENP-N$^{NT}$ constructs in buffer containing 20 mM Tris.Cl, pH 7.5, 1 mM EDTA, 150 mM NaCl, 1 mM DTT and 0.1% CHAPS for 30 min at room temperature. The mixture was analyzed by 5% native PAGE, and the gel was stained with SYBR Gold. The gel was imaged at 488 nm (for SYBR Gold staining or Alexa488 labelled H2B in the nucleosome) and 647 nm (for Atto

N 647 labeled H2B in the nucleosome) using a Typhoon imager (GE Healthcare). Intensity at 647 nm was used for binding curves, with the number of replicates indicated. The intensity of the nucleosome band (not the shifted bands) was quantified, and normalized to the nucleosome sample without CENP-N.

## Isothermal titration calorimetry

All protein samples were loaded onto a Superdex 200 10/300 column equilibrated in 20 mM HEPES pH 7.5, 2.5% glycerol, 300 mM NaCl and 1 mM TCEP prior to ITC runs. ITC measurements were performed at 25°C on an ITC200 micro calorimeter (GE Healthcare). In each titration, the protein in the cell (at a 5–8 µM concentration) was titrated with $19 \times 2$ µl injections (at 180 s intervals) of protein ligand (at 50–80 µM concentration). The injections were continued beyond saturation to allow for determination of heats of ligand dilution. Data were fitted by least-square procedure to a single site binding model using ORIGIN software package (MicroCal, Malvern Instruments, Worcestershire, UK).

## Mammalian plasmids

Plasmids for stable cell lines were generated in pCDNA5/FRT/TO-EGFP-IRES, a modified version of the pCDNA5/FRT/TO vector (Invitrogen, Carlsbad, CA). The starting plasmid for EGFP expression was made by PCR amplifying the EGFP sequence of the pEGFP-C1 plasmid (Takara Bio Inc., Shiga, Japan) and cloning it into the pcDNA5/FRT/TO vector previously modified to carry an internal ribosomal entry site (IRES) sequence creating the pcDNA5/FRT/TO EGFP-IRES vector (*Petrovic et al., 2010*). Mammalian expression plasmids used in this study to express N-terminally tagged CENP-C WT or the PEST mutant were derived from the pCDNA5/FRT/TO-EGFP-IRES and used either for genomic integration or transient expression of human CENP-C proteins. A modified version of the pCDNA5/FRT/TO-EGFP-IRES plasmid where EGFP was removed and replaced with mCherry to create the pCDNA5/FRT/TO-mCherry-IRES was used to express all C-terminally tagged CENP-N constructs either via genomic integration or transient expression. To create the EGFP tagged CENP-C protein and CENP-N tagged with mCherry protein full-length proteins were amplified by PCR from full-length human codon-optimized cDNA synthesized by GeneArt (Life Technologies, Thermo Fisher, USA). Using the Gibson cloning method[36] the cDNA sequence was ligated with the PCR amplified pCDNA5/FRT/TO-EGFP-IRES or pCDNA5/FRT/TO-mCherry-IRES to create the final constructs used in this study. Each plasmid was then sequence verified before use.

## Cell culture and transfection

U2OS cells, a gift from A. Bird (MPI-Dortmund, Germany), were grown in DMEM (PAN Biotech, Aidenbach, Germany) supplemented with 10% FBS (Clontech, part of Takara Bio group, Shiga, Japan), penicillin and streptomycin (GIBCO, Carlsbad, CA), and 2 mM L-glutamine (PAN Biotech). Transient transfections were performed using pCDNA5-FRT-TO plasmids carrying either CENP-N$^{wt}$ or mutants with a C-terminal-mCherry tag transfected into asynchronously growing cells and expressed for 24 hr before preparation for immunofluorescence analysis. To analyze whether the CENP-C$^{5A}$ mutant abrogates kinetochore localization of CENP-N, we used Flp-In T-REx HeLa cells (a gift from SS Taylor, University of Manchester, Manchester, England, UK) to generate stable doxycycline-inducible cell lines, which were maintained in DMEM (PAN Biotech, Aidenbach, Germany) with 10% tetracycline-free FBS (Clontech) supplemented with 50 µg/ml Zeocin (Invitrogen) and 2 mM L-glutamine (PAN Biotech). Flp-In T-REx HeLa expression cell lines were generated as previously described (*Krenn et al., 2012*). FlpIn T-REx HeLa cells stably expressing mCherry-CENP-N were grown on coverslips pre-coated with poly-D-Lysine (Millipore, 15 µg/ml). The day after cells were seeded, expression of mCherry-CENP-N was induced using 0.2–0.5 µg/ml doxycycline (Sigma, St. Louis, MO). Further, cells were transiently transfected with GFP-CENP-C$^{WT}$ or GFP-CENP-C$^{5A}$ mutant and CENP-C siRNA for 72 hr. When required 16 hr before fixation cells were synchronized overnight in 0.33 µM nocodazole (Sigma–Aldrich) to assess mitotic localization. Endogenous CENP-C was depleted by siRNA (target sequence: 5'-GGAUCAUCUCAGAAUAGAA-3' obtained from ThermoFisher), transfected into cells using Lipofectamine RNAiMAX Transfection Reagent (ThermoFisher) as per the manufactures instructions for 72 hr.

## Immunofluorescence

Either U2OS or FlpIn T-REx HeLa cells were fixed using 4% paraformaldehyde in phosphate buffered saline (PBS), permeablised using 0.25% Triton X-100 in PBS and blocked in 3% BSA/PBS. U2OS cells were stained for with the following antibodies: anti-CENP-C [rabbit polyclonal antibody SI410, diluted 1:1000 (*Trazzi et al., 2009*); and CREST/anti-centromere antibody (Antibodies Inc. 15-234-0001), diluted 1:100]. FlpIn T-REx HeLa cells were stained for GFP (GFP-Boost, Chromotek gba-488, 1:500), mCherry (RFP-Boost, Chromotek, rba-594, 1:500), CENP-C [rabbit polyclonal antibody SI410; 1:1000 (*Trazzi et al., 2009*), CREST/anti-centromere antibodies (Antibodies, Inc., Davis, CA, 1:100), or CENP-A (Mouse, Gene Tex GTX13939, 1:500) diluted in blocking buffer for 2–4 hr. Donkey anti-human Alexa Fluor 405 and donkey anti-rabbit Alexa Fluor 488, donkey anti-human (Jackson ImmunoResearch Laboratories, Inc., West Grove, PA), as well as donkey anti-mouse (Invitrogen) were used as secondary antibodies. DNA was stained with 0.5 µg/ml DAPI (Serva), and coverslips were mounted with Mowiol mounting media (Calbiochem). U2OS cells were imaged with a Deltavision Elite System (GE Healthcare, UK) equipped with an IX-71 inverted microscope (Olympus, Japan), a PLAPON 60x/1.42NA objective and a pco.edge sCMOS camera (PCO-TECH Inc., USA). Images were acquired as Z-sections (using the softWoRx software from Deltavision) and converted into maximal intensity projections TIFF files for illustrative purposes while FlpIn T-REx HeLa Cells were imaged at room temperature using a spinning disk confocal device on the 3i Marianas system equipped with an Axio Observer Z1 microscope (Zeiss), a CSU-X1 confocal scanner unit (Yokogawa Electric Corporation, Tokyo, Japan), Plan-Apochromat 63 × or 100×/1.4NA Oil Objectives (Zeiss), and Orca Flash 4.0 sCMOS Camera (Hamamatsu). Images were acquired as z-sections at 0.2 µm. Images were converted into maximal intensity projections, exported, and converted into 8-bit. Quantification of kinetochore signals was performed on unmodified 16-bit z-series images using Imaris 7.3.4 32-bit software (Bitplane, Zurich, Switzerland). Measurements were exported in Excel (Microsoft) and graphed with GraphPad Prism 6.0 (GraphPad Software, San Diego California USA). For quantification in *Figure 4B-C* after background subtraction, all signals were normalized to CREST and the values obtained were then normalized by the mean of the CENP-N-mCherryWT construct. Quantifications are based on one experiment where a range of 7 to 10 cells and 177 to 393 kinetochores per condition were analyzed.

## Crystallization of CENP-N$^{1-235}$his and structure determination

Initial crystallization hits of CENP-N$^{1-212}$His6 (His6, hexahistidine tag) or CENP-N$^{1-235}$His6 were obtained in sitting drop crystallization experiments at c.a. 6 mg/ml in a 96 well format using a Mosquito protein crystallization robot (TTP Labtech) at 4°C. Crystals grew in various commercial screens including Qiagen Nextal PEGSII conditions B11 and B12 and Qiagen Nextal PEG conditions H6 and H8 within 24–48 hr, reaching maximum size in 5–7 days. CENP-N$^{1-235}$His6 crystals were further optimized in 24-well plates via hanging drop method using a two-dimensional grid screen varying PEG3350 (from 6–16%) and pH (from 6.6 to 7.2). Selenomethionine (SeMet) derivatives of CENP-N$^{1-235}$His6 were crystallized under similar conditions. Crystals were cryo-cooled in a mother liquor solution containing 20–25% (v/v) glycerol. All data were collected at 100K using a Pilatus 6M detector either at the X10SA beamline at the SLS in Villigen, Switzerland, or at the P11 beamline of PETRA in Hamburg, Germany. All data sets were integrated and scaled using XDS and XSCALE (*Kabsch, 2010*).

Both native and selenomethionine (SeMet) crystals grew in space group P4$_1$ with two molecules per asymmetric unit and a relatively similar packing, although the c-axis is approximately 4 Å shorter in the latter. Phasing with PHENIX (*Adams et al., 2010*) located 12 of the 14 possible SeMet sites (in each monomer a C-terminal Met is disordered, and M167 has very weak anomalous density). Merging of SeMet datasets from two different crystals was essential to improve the anomalous signal. The quality of the phases allowed autobuilding of (mostly) alpha helices into the electron density in spite of the relatively low resolution (3.3 Å). Molecular replacement with PHASER (*Collaborative Computational Project, Number 4, 1994*) successfully placed this initial model into the native dataset (conservative resolution at I/sigma = 3 of 2.89 Å, data used to 2.74 Å). The sequence was assigned with the help of the anomalous peaks. Refinement with REFMAC (*Collaborative Computational Project, Number 4, 1994*) and PHENIX resulted in a model with very good Ramachandran geometry (98% residues in favoured regions, 0% outliers) and R$_{work}$/R$_{free}$ values

of 21.6% and 26.0%, respectively. Data to 2.74 Å were used despite high R factors, since they improved the convergence and quality of the refinement.

Monomers A and B in the asymmetric unit are quite similar except for a minor hinge-bending between the two subdomains of each monomer, and except for loops 137–142 and 164–174. In monomer B, the 137–142 loop is pulled approximately 3 Å away from the remainder of the molecule by a symmetry contact, and in turn pulls on the neighbouring loop causing a 2.6 Å movement of the backbone of K117. This loop has no crystal contacts in monomer A. Similarly, residues 164–174 of monomer B pack against a symmetry related molecule, most likely causing the observed (relative) stabilization of M167 in monomer B and backbone shifts of up to 7.3 Å relative to monomer A. Residues 166–168 have very weak density especially in chain A, corresponding to a weak and multiple anomalous density of SeM167, indicating multiple conformations of this loop. The coordinates of CENP-N have been submitted to the protein data bank (PDB) with code 6EQT.

## CryoEM grid preparation and microscopy

Quantifoil 2/2 grids (Quantifoil Micro Tools GmbH, Grossloebichau, DE) were used for the CENP-$N^{NT}$:CENP-$A^{NCP}$ complex (2.5 µM concentration). The grids were glow discharged (EMItec, Lohmar, DE) at 40 mA for 20 s. 4 µl sample was applied onto the grid before plunge freezing into Ethane, using a Vitrobot (FEI, MARK IV). Blot time was 4 s. All grids were stored in liquid nitrogen before imaging. CENP-$N^{NT}$:CENP-$A^{NCP}$ complex was imaged at nominal magnification of 29000x on a FEI Titan Krios (300 kV), equipped with a Gatan K2 Summit direct detector. Pixel size was 1.02 Å. The movies were captured in super resolution mode with electron dose rate at 10 electrons per pixel per second for 8 s and 0.2 s per frame. Defocus range was −1.0 to −2.5 µm.

## Single particle analysis of cryoEM images

Motioncor2 was used for the alignment of images (motion correction) (*Zheng et al., 2017*). GCTF was applied for Constant transfer function (CTF) estimation (*Zhang, 2016*). Images were evaluated manually by inspecting their power spectra. Particles were manually picked for the initial 2D classification (10 class averages) in RELION 2.05 (*Fernandez-Leiro and Scheres, 2017*). Initial 2D class averages were used for particle auto-picking as described in Relion2.0 tutorials. Particles from auto-picking were extracted and sorted for reference free 2D classification. In 2D classification, 200 class averages were generated. Noisy class averages were discarded. Particles from the retained class averages were used for reference free 3D reconstruction in cryoSPARC (*Punjani et al., 2017*). The low-pass filtered map from the 3D reconstruction in cryoSPARC was used as reference for 3D classification in RELION 2.05 (*Fernandez-Leiro and Scheres, 2017*). Four 3D classes were created after 3D classification. Particle images from the class at high resolution were used for 3D refinement in Relion 2.1b1. The map was then sharpened in 'post-process', using a mask file created by Relion2.1b1. The local resolution of the map was estimated by Relion (*Kucukelbir et al., 2014*). Anisotropy was analyzed by 3DFSC calculation (*Tan et al., 2017*).

## CryoEM structure modeling

The original 3D refined map and post-processed map were used for the model fitting and refinement. DNA was taken from a high-resolution crystal structure of a nucleosome with 601 DNA [Protein Data Bank accession code 3LZ0 (*Vasudevan et al., 2010*), CENP-A containing histone core was taken from Protein Data Bank accession code 3AN2 (*Tachiwana et al., 2011*). The crystal structure of CENP-N (amino acids 1–212) was used to fit into the remaining density using UCSF Chimera (*Pettersen et al., 2004*) (rigid body without allowing flexibility). Based on map density, the model was iteratively modified and locally refined in Coot (*Emsley et al., 2010*). The final model was subjected to real space refinement in PHENIX (*Adams et al., 2010*). The coordinates and maps were deposited in the protein data base (pdb code 6C0W, and EMD-7326).

## Acknowledgements

We thank Chuan Hong, Rick Huang, and Zhiheng Yu at the HHMI Janelia Farm CryoEM Facility for help in microscope operation and data collection, Bridget Carragher, Clint Potter and Zhening Zhang at NRAMM for data collection and advice, staff at the University of Colorado Boulder EM Services Core Facility, the beamline staff of X10SA at the Swiss Light Source Paul Scherrer Institute,

Villigen (Switzerland), and of P11 at the PETRA synchrotron, Hamburg (Germany), for support, and our colleagues of MPI Dortmund for help with the data collection, Christos Gatsogiannis for advice on the EM data, Kerstin Klare for initial contributions to the project, and Doro Vogt, Annika Take, and Sabine Wohlgemuth for excellent technical assistance. AM acknowledges funding by the the Horizon 2020 ERC agreement RECEPIANCE (AdG 669686), and the DFG's Collaborative Research Centre (CRC) 1093. KL received funding from NIH (GM067777), and from the Howard Hughes Medical Institute. Some of this work was performed at the National Resource for Automated Molecular Microscopy located at the New York Structural Biology Center, supported by grants from the NIH National Institute of General Medical Sciences (GM103310) and the Simons Foundation (349247). The authors declare no competing financial interests.

## Additional information

### Competing interests
Andrea Musacchio: Senior Editor, *eLife*. The other authors declare that no competing interests exist.

### Funding

| Funder | Grant reference number | Author |
| --- | --- | --- |
| H2020 European Research Council | AdG 669686 | Andrea Musacchio |
| Deutsche Forschungsgemeinschaft | CRC1093 | Andrea Musacchio |
| National Institutes of Health | GM067777 | Karolin Luger |
| Howard Hughes Medical Institute | | Karolin Luger |
| Max-Planck-Gesellschaft | Open-access funding | Andrea Musacchio |

The funders had no role in study design, data collection and interpretation, or the decision to submit the work for publication.

### Author contributions
Satyakrishna Pentakota, Keda Zhou, Validation, Investigation, Methodology, Writing—review and editing; Charlotte Smith, Investigation; Stefano Maffini, Formal analysis, Methodology; Arsen Petrovic, Investigation, Methodology; Garry P Morgan, Methodology; John R Weir, Supervision; Ingrid R Vetter, Investigation, Methodology, Writing—review and editing; Andrea Musacchio, Conceptualization, Supervision, Funding acquisition, Writing—original draft, Project administration; Karolin Luger, Conceptualization, Supervision, Funding acquisition, Writing—review and editing

### Author ORCIDs
Keda Zhou http://orcid.org/0000-0003-0258-1489
Andrea Musacchio http://orcid.org/0000-0003-2362-8784
Karolin Luger http://orcid.org/0000-0001-5136-5331

### Decision letter and Author response
Decision letter https://doi.org/10.7554/eLife.33442.030
Author response https://doi.org/10.7554/eLife.33442.031

## Additional files

### Supplementary files
• Transparent reporting form
DOI: https://doi.org/10.7554/eLife.33442.024

## Major datasets

The following datasets were generated:

| Author(s) | Year | Dataset title | Dataset URL | Database, license, and accessibility information |
|---|---|---|---|---|
| Pentakota S, Keda Zhou, Stefano Maffini, Arsen Petrovic, Garry P Morgan, Charlotte Smith, John R Weir, Ingrid R Vetter, Andrea Musacchio, Karolin Luger | 2017 | Crystal structure of the human kinetochore protein CENP-N | http://www.rcsb.org/pdb/search/structid-Search.do?structureId=6EQT | Publicly available at the RCSB Protein Data Bank (accession no. 6EQT) |
| Zhou K, Pentakota S, Vetter IR, Morgan GP, Petrovic A, Musacchio A, Luger K | 2017 | Cryo-EMstructure of human kinetochore protein CENP-N with the centromeric nucleosome containing CENP-A | http://www.rcsb.org/pdb/search/structid-Search.do?structureId=6C0W | Publicly available at the RCSB Protein Data Bank (accession no. 6C0W) |

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
