## [Decision Letter]

Thank you for submitting your article "Decoding the centromeric nucleosome through CENP-N" for consideration by *eLife*. Your article has been reviewed by three peer reviewers, one of whom, Geeta J Narlikar (Reviewer #1), is a member of our Board of Reviewing Editors, and the evaluation has been overseen by John Kuriyan as the Senior Editor. The following individual involved in review of your submission has agreed to reveal their identity: Hongtao Yu (Reviewer #2).

The reviewers have discussed the reviews with one another and the Reviewing Editor has drafted this decision to help you prepare a revised submission. As you will see the reviewers agree that your work is a significant advance in understanding the structural basis for how kinetochores are assembled on centromeres. They have some comments that we would like you to address. We anticipate that you can address these comments with either textual clarifications or re-analysis of some of the existing data.

Summary

Functional centromeres are epigenetically marked by the histone H3 variant CENP-A and form the foundation for the assembly of the kinetochore, which provides landing pads for spindle microtubules and spindle checkpoint proteins. Proper assembly of the kinetochore is thus critical for accurate chromosome segregation and genomic stability. The constitutive centromere associated network of proteins (CCAN) directly recognizes CENP-A-containing nucleosomes and builds the inner kinetochore. Two CCAN components, CENP-C and CENP-N, have been shown to specifically and cooperatively bind CENP-A nucleosomes. The structure of the CENP-A nucleosome and of a CENP-C fragment bound to the CENP-A nucleosome has been reported previously. CENP-N was the first CCAN protein identified that selectively binds CENP-A nucleosomes in chromatin. How CENP-N achieved this selective binding has been a mystery until this work. Further how CENP-N and CENP-C cooperate to specifically recognize CENP-A nucleosomes was also not known.

This manuscript makes 3 major contributions: 1) It provides the X-ray crystallographic structure of the CENP-A nucleosome binding region of CENP-N; this allowed the authors to assign two structural domains (the PYRIN domain and the CLN-HD) which could not be predicted from sequence alone. 2) It provides the first insight into how CENP-N binds to CENP-A nucleosomes by solving a Cryo-EM structure of CENP-N bound to the CENP-A nucleosome. While there was indication that the L1 loop region of CENP-A was involved in the interaction, this work shows how CENP-N engages the nucleosome and also discovers a surprisingly large DNA-interacting domain. This binding mode suggest that CENP-N can antagonize the nucleosome destabilizing action of chromatin remodelers. 3) This work furthers our understanding of the interplay between CCAN proteins, in particular interaction between the CENP-L/N complex and CENP-C through a conserved CENP-C motif, and how their interactions contribute to centromere assembly.

This work provides a structural explanation of why some previously reported mutations in CENP-N (R11A, K15A, K45A) break the interaction of CENP-N with the CENP-A nucleosome and fail to target CENP-N to centromeres. The structural studies are supported by in vitro biochemistry to probe the interactions and cell biology experiments to test their relevance in vivo. This work also brings up many interesting questions that are beyond the scope of this manuscript. With the identification of the PYRIN domain, does CENP-N interact with itself or other PYRIN domain containing proteins since PYRIN domains have been found to be protein-protein interaction domains? Is CENP-N binding affecting the conformation of DNA ends in CENP-A NCPs and if so, how? With the ability to potentially separate the functions of CENP-C (i.e. HIKM binding, CENP-A binding, CENP-NL binding), how do these different 'activities' contribute to overall centromere assembly and function? Overall the findings and new questions that are raised greatly advance the molecular understanding of centromere and kinetochore assembly.

Essential revisions:

The authors perform mutational analysis to functionally validate their structural information. In general, the trends support the structural insights since mutating a few of the interacting residues caused at least some defect in either pull down assays or localization assays. However, the following points should be further clarified.

1) A naïve assumption is that CENP-N localization to centromeres would depend on its ability to selectively recognize CENP-ANCPs over H3NCPs. However, while the authors see that a CENP-N E3A/E7A mutant weakens CENP-N's affinity for CENP-ANCPs, oddly more so than its affinity for H3NCPs, they do not examine the localization of this mutant. This data is important to confirm the validity of nature of the CENP-N – CENP-ANCPs specific interaction suggested by the structures. Can the authors clarify why they did not look at localization of the CENP-N E3A/E7A mutant? In addition, quantification and statistical testing of Figure 4—figure supplement 2 would help to give some confidence that this idea is correct in cells.

In Figure 4 the authors focus on the R11A mutant based where they say: "In pull-down assays in vitro, we found essentially complete loss of binding with an alanine (A) mutant of Arg11 (R11), and substantial reductions of binding with alanine mutants of E7, K15, K45, and Y147 (Figure 4—figure supplement 2)." The pull down data for the R11A mutant does not appear to be in that figure and should be added.

2) There is some concern that normalizing to CREST staining in Figure 4 could confuse the results as CREST serum recognizes some mixture of CENP-A, B, C proteins. Because others have shown that CENP-A and CENP-C levels at centromeres may depend on CENP-N, the mCherry intensities alone should be quantified as these would be a more direct assessment.

3) Related to point 2, showing that expression levels of the different mutants are similar would allay the concern that the localization differences are simply due to expression level differences.

4) The largest decrease in fraction bound in Figure 4—figure supplement 2 was the K143A/Y147A mutant. Why was K143A selected? What are its effects in the other assays? There other places where the logic of why certain mutants were chosen over others is lacking. Thus why not E3A since E3 and E7 were presented as two crucial factors in both the text and structural data in differentiating between CENP-A and H3 nucleosomes?

---

## [Author Response]

Essential revisions:The authors perform mutational analysis to functionally validate their structural information. In general, the trends support the structural insights since mutating a few of the interacting residues caused at least some defect in either pull down assays or localization assays. However, the following points should be further clarified.1) A naïve assumption is that CENP-N localization to centromeres would depend on its ability to selectively recognize CENP-ANCPs over H3NCPs. However, while the authors see that a CENP-N E3A/E7A mutant weakens CENP-N's affinity for CENP-ANCPs, oddly more so than its affinity for H3NCPs, they do not examine the localization of this mutant. This data is important to confirm the validity of nature of the CENP-N – CENP-ANCPs specific interaction suggested by the structures. Can the authors clarify why they did not look at localization of the CENP-N E3A/E7A mutant? In addition, quantification and statistical testing of Figure 4—figure supplement 2 would help to give some confidence that this idea is correct in cells.

We agree with the reviewers that this point needed clarification. We have now added new EMSA experiments in a new Figure 4—figure supplement 1 that will considerably facilitate the interpretation of the effects of mutations (this figure replaces the EMSA assays in panel B of Figure 4—figure supplement 2 as the new assays include the same experiments, together with additional mutants). Specifically, we have now included the R11A mutant in the same type of EMSA assay in which we analyze the E3A-E7A mutant. Importantly, R11A now appears consistently in all our assays, including solid phase, kinetochore localization, and EMSA, thus providing a clear beacon to interpret the behavior of the other mutants.

In the new Figure 4—figure supplement 1, we show that R11A behaves like E3A-E7A, i.e. it decreases the affinity for CENP-A^NCPs^ but not for H3^NCPs^, like all other CENP-N mutants at the interface with CENP-A. This is expected: CENP-N has an extensive interface with DNA, and in the EMSA assays at low salt this triggers measurable binding also to H3^NCPs^. We show that this “background” binding to H3^NCPs^, contrarily to binding to CENP-A^NCPs^, is not affected by mutations targeting the CENP-A binding interface of CENP-N. This explains why these mutations (which were deliberately chosen to only affect histone recognition, but not DNA interactions) affect CENP-A^NCP^ binding more than they affect H3^NCPs^ binding, and we now clarify this in the text.

We did not include the E3A-E7A mutant in our collection of mutants studied in vivobecause the time constraints imposed by the competing paper made it difficult to harmonize the diverse sets of mutants that were generated in the two collaborating labs. We believe that demonstrating the effects of all mutants in all assays will not contribute to the conclusions of our manuscript, as these experiments collectively, and conclusively, validate the nucleosome-CenpN interfaces determined by cryo-EM.

In Figure 4 the authors focus on the R11A mutant based where they say: "In pull-down assays in vitro, we found essentially complete loss of binding with an alanine (A) mutant of Arg11 (R11), and substantial reductions of binding with alanine mutants of E7, K15, K45, and Y147 (Figure 4—figure supplement 2)." The pull down data for the R11A mutant does not appear to be in that figure and should be added.

We apologize for the confusion. The pull down data for the Arg11 mutant was already shown in Figure 4, we have now referred more clearly to this panel.

2) There is some concern that normalizing to CREST staining in Figure 4 could confuse the results as CREST serum recognizes some mixture of CENP-A, B, C proteins. Because others have shown that CENP-A and CENP-C levels at centromeres may depend on CENP-N, the mCherry intensities alone should be quantified as these would be a more direct assessment.

We agree that this might have been a concern, had we been working under conditions of RNAi of endogenous CENP-N. However, the experiment was carried out in presence of endogenous CENP-N, i.e. we were only measuring the relative ability of mCherry-labeled wild type and mCherry-labeled mutant constructs to reach the kinetochore in presence of endogenous CENP-N. Under these conditions, we do not expect any loss of CENP-C or other antigens recognized by the CREST serum, as endogenous CENP-N is available to “supplement” a requirement for this protein even if the mutants cannot reach the kinetochore.

3) Related to point 2, showing that expression levels of the different mutants are similar would allay the concern that the localization differences are simply due to expression level differences.

Because of time constraints, we were not able to include a western blot to demonstrate with a single gel that the expression levels of mutants tested in the localization studies in cells are similar, (see also point below). However, we are confident that the expression levels are similar based on individual mutant plasmid titration experiments. We now write that the expression of mutants was similar and refer to these data as “unpublished data”

4) The largest decrease in fraction bound in Figure 4—figure supplement 2 was the K143A/Y147A mutant. Why was K143A selected? What are its effects in the other assays? There other places where the logic of why certain mutants were chosen over others is lacking. Thus why not E3A since E3 and E7 were presented as two crucial factors in both the text and structural data in differentiating between CENP-A and H3 nucleosomes?

K143 was selected because it is positioned at the interface with the CENP-A L1 loop. We now indicate this more clearly in the text. We agree with the reviewers that the choice of mutants is less logical than one might have hoped, which, as explained above, was in part due to the fact that different sets of mutants were generated independently in the two collaborating labs. We note however that the situation is not as bad as it might seem. However, to reconcile our datasets, we have now created a “standard” with the R11A mutant, which is used across all assays. Second, we show through the EMSA assay that all mutants that affect the direct interaction of CENP-N with the CENP-A L1 loop cause a decrease in binding selectivity for CENP-A, as expected. Finally, we show that mutations at the interface with the L1 loop, alone or in combination with mutation in the DNA binding region of CENP-N, cause displacement from the kinetochore. We believe that overall this offers a clear picture of the effects of mutations on various functional interfaces of CENP-N, and present a strong validation for the identified interfaces.